



# FLEXPART version 11: Improved accuracy, efficiency, and flexibility

Lucie Bakels[1], Daria Tatsii[1], Anne Tipka[1,3], Rona Thompson[5], Marina Dütsch[1], Michael Blaschek[1], Petra Seibert[1,2], Katharina Baier[1], Silvia Bucci[1], Massimo Cassiani[5], Sabine Eckhardt[5], Christine Groot Zwaaftink[5], Stephan Henne[4], Pirmin Kaufmann[7], Vincent Lechner[1], Christian Maurer[6], Marie D. Mulder[6], Ignacio Pisso[5], Andreas Plach[1], Rakesh Subramanian[1], Martin Vojta[1], and Andreas Stohl[1]

[1]Department of Meteorology and Geophysics, University of Vienna, Vienna, Austria
[2]Institute of Meteorology and Climatology, University of Natural Resources and Life Sciences (BOKU), Vienna, Austria
[3]Preparatory Commission for the Comprehensive Nuclear-Test-Ban Treaty Organisation (CTBTO), Vienna, Austria
[4]Swiss Federal Laboratories for Materials Science and Technology (EMPA), Zurich, Switzerland
[5]NILU, Kjeller, Norway
[6]GeoSphere Austria, Vienna, Austria
[7]Federal Office of Meteorology and Climatology MeteoSwiss, Zurich Airport, Switzerland

**Correspondence:** Lucie Bakels (lucie.bakels@univie.ac.at)

**Abstract.** Numerical methods and simulation codes are essential for the advancement of our understanding of complex atmospheric processes. As technology and computer hardware continue to evolve, the development of sophisticated code is vital for accurate and efficient simulations. In this paper, we present the recent advancements made in FLEXPART, a Lagrangian particle dispersion model, which has been used in a wide range of atmospheric transport studies over the past three decades,

extending from tracing radionuclides from the Fukushima nuclear disaster, to inverse modelling of greenhouse gases, and to the study of atmospheric moisture cycles.

This version of FLEXPART includes notable improvements in accuracy and computational efficiency. 1) By leveraging the native vertical coordinates of European Centre for Medium Range Weather Forecasts (ECMWF) Integrated Forecasting System (IFS) instead of interpolating to terrain-following coordinates, we achieved an improvement in trajectory accuracy, leading to

a $\sim 8 - 10$ % reduction in conservation errors for semi-conserved quantities like potential vorticity. 2) The shape of aerosol particles is now accounted for in the gravitational settling and dry deposition calculation, increasing the simulation accuracy for non-spherical aerosol particles such as microplastic fibers. 3) Wet deposition has been improved by the introduction of a new below-cloud scheme, a new cloud identification scheme, and by improving the interpolation of precipitation. 4) Functionality from a separate version of FLEXPART, the FLEXPART-CTM model, is implemented, which includes linear chemical reactions.

Additionally, the incorporation of Open Multi-Processing parallelisation makes the model better suited for handling large input data. Furthermore, we introduced novel methods for the input and output of particle properties and distributions. Users now have the option to run FLEXPART with more flexible particle input data, providing greater adaptability for specific research scenarios (e. g., effective backward simulations corresponding to satellite retrievals). Finally, a new user manual (https://flexpart.img.univie.ac.at/docs/) and restructuring of the source code into modules will serve as a basis for further development.



## 1 Introduction

Atmospheric transport modelling plays an important role in understanding many of the complex interactions within our atmosphere. Traditionally, Eulerian methods have been widely used for such modelling. These methods solve the atmospheric advection-diffusion equation on a fixed grid and give the concentration of tracer gases or aerosols at each grid point as a function of time. The advantage of Eulerian methods is that they can take into account all processes occurring during transport including nonlinear atmospheric chemistry. However, they typically have a relatively high level of numerical diffusion (e. g., Reithmeier and Sausen, 2002; Cassiani et al., 2016) limiting their capability of preserving plumes over long transport distances (Rastigejev et al., 2010) and the spatial resolution of the concentration fields is limited by the grid spacing of the model. Lagrangian methods, on the other hand, follow individual 'computational particles' (from now on referred to as 'particle') and simulate their transport based on large-scale winds from Eulerian input data and parameterised small-scale fluctuations. They are independent of the computational grid and can therefore produce concentration fields with potentially infinitesimal spatial resolution at relatively low computational cost, especially if several tracers are to be transported simultaneously. Another advantage of Lagrangian methods is that they can provide a direct link between regions along particle trajectories.

Atmospheric transport models using the Lagrangian method with parameterisation of sub-grid processes are typically referred to as Lagrangian Particle Dispersion Models (LPDMs). One prominent LPDM is the FLEXible PARTicle model (FLEXPART), which was developed nearly three decades ago to simulate the dispersion of radionuclides resulting from nuclear disasters such as the Chernobyl accident (Stohl et al., 1998a). Since its inception, FLEXPART has proven to be a valuable tool for studying a wide range of environmental problems, both in research and operational settings. Its applications now cover large parts of atmospheric research, including the simulation of the transport of heat and water in the atmosphere (e. g., Baier et al., 2022; Peng et al., 2022), volcanic and wildfire plumes (e. g., Stohl et al., 2006, 2011; Moxnes et al., 2014), transport and fall-out after nuclear accidents or explosions (e. g., Stohl et al., 2012; Arnold et al., 2015), transport of aerosols such as dust (e. g., Groot Zwaaftink et al., 2017; Ryder et al., 2019) and microplastics (e. g., Evangelou et al., 2024; Evangeliou et al., 2020), the interpretation of biogenic secondary organic aerosol compound measurements (e. g., Martinsson et al., 2017), the transport of pollutants into remote regions like the Arctic (e. g., Dada et al., 2022; Zhu et al., 2020), the interpretation of ice cores (e. g., Eckhardt et al., 2023; McConnell et al., 2024), and the modelling of emission sensitivities for greenhouse gas atmospheric concentrations (e. g., Bergamaschi et al., 2022; Vojta et al., 2022).

Several other LPDMs for regional and large-scale atmospheric transport modelling exist, e. g., HYSPLIT (Draxler and Hess, 1998), STILT (Lin et al., 2003), TRACMASS (Döös et al., 2017), MPTRAC (Hoffmann et al., 2022), and ATTILA (Brinkop and Jöckel, 2019). Nevertheless, FLEXPART combines a unique set of capabilities no other model can offer, including (i) detailed parameterisations for Gaussian (Stohl and Thomson, 1999) and non-Gaussian (Cassiani et al., 2015) turbulence in the atmospheric boundary layer (ABL), which both take into account the vertical gradient of air density; (ii) a subgrid convection parameterisation (Forster et al., 2007); (iii) treatment of radioactive decay in the atmosphere and on the ground (Stohl et al., 1998a); (iv) detailed parameterisations for dry and wet deposition and gravitational settling (Stohl et al., 2005); (v) linear chemical reaction with hydroxyl radicals (Pisso et al., 2019); (vi) the capability of running the model backward in time to



create sensitivities of atmospheric concentrations and depositions to emission sources (Seibert and Frank, 2004; Eckhardt
et al., 2017) that can be interfaced directly with an inverse modelling code for determining emissions (Thompson and Stohl,
2014); and (vii) a domain-filling option where the entire atmosphere can be filled with particles (Stohl and James, 2004), which
is useful for producing Lagrangian climate diagnostics (e. g., Schicker et al., 2010; Baier et al., 2022) and three-dimensional
concentration fields of greenhouse gases that may serve as initial conditions for inverse modelling (Groot Zwaaftink et al.,
2018; Vojta et al., 2022).

This paper describes a new model version, FLEXPART 11, that adds four important new features to FLEXPART's capabilities as well as other improvements. Firstly, when driven with data from the European Centre for Medium Range Weather
Forecasts (ECMWF), FLEXPART 11 offers the option to keep the original model layers intact for the transport with resolved-scale winds, instead of interpolating to a coordinate system with fixed heights above terrain as previous versions did (see
section 3). We show that this improves the overall accuracy of FLEXPART model simulations. Secondly, FLEXPART 11 can
calculate the drag coefficient of non-spherical aerosol particles of various shapes, which improves the calculated gravitational
settling of such particles substantially and thereby has a large effect on their overall simulated transport in the atmosphere (see
section 4). Thirdly, a new below-cloud scavenging scheme for aerosols is introduced, as well as an improved interpolation
scheme for precipitation and a new cloud layer identification scheme (see section 5). Lastly, linear chemical reactions are implemented based on the initial work of Henne et al. (2018) who developed the FLEXPART-CTM model from FLEXPART 8
(where CTM stands for chemical transport model), which was first described in Groot Zwaaftink et al. (2018) (see section 6). In
addition to enhancing the model's core functionality, we have improved the user interface by incorporating additional options
for ways of running FLEXPART (see section 9).

Computational efficiency is another important consideration. Legacy models such as earlier versions of FLEXPART or HYS-PLIT (Draxler and Hess, 1998) were originally designed for serial processing. An important step was made with FLEXPART
10.4 which introduced the option to use MPI parallelisation (Pisso et al., 2019). However, especially with increasing size of
the meteorological input data, the memory requirement became problematic in parallel mode. For this reason, we opted for a
different parallelisation strategy, using OpenMP (Open Multi-Processing), following FLEXPART-CTM (Henne et al., 2018)
(see section 8). Furthermore, numerous features added over a period of more than 25 years and often deviating from the original coding standards make the FLEXPART 10.4 source code difficult to understand and maintain. Models that were created
recently such as MPTRAC (Hoffmann et al., 2022) are designed from scratch to run on current hardware, for instance by
utilizing the acceleration offered by graphics processing units (GPUs), while not being under the constraint of a legacy code
base. However, as FLEXPART is a model that is well-tested for a wide range of applications, offers many features not available
in other models, and has a large user community, a complete rewrite of the code did not seem appealing. Instead, the approach
that started with FLEXPART 8 and 10 to introduce features of modern Fortran standards into the source code was continued.
In FLEXPART 11, all programme subunits have been encapsulated into modules and more use has been made of whole-array
syntax.



We validate our changes by comparing FLEXPART 11 to tracer experiments (section 7). We also test the conservation of meteorological tracers (section 3.2), the degree to which particles in a global domain-filling simulation stay well mixed (section 3.3), and the removal processes by reproducing the Fukushima Dai-Ichi nuclear power plant accident (section 5.4).

For users unfamiliar with FLEXPART, a short overview of FLEXPART 11 capabilities, input data, and code reorganisation can be found in the supplement (see section A). Accompanying this paper is a completely revised technical documentation of FLEXPART (https://flexpart.img.univie.ac.at/docs); the code is available from https://gitlab.phaidra.org/flexpart/flexpart. We recommend users of FLEXPART to consult this living document for updates and future code developments.

## 2    Input data

FLEXPART calculates particle trajectories using interpolated meteorological fields, e. g., grid-scale three-dimensional fields of wind velocities, density, temperature, specific humidity, cloud liquid and ice water content, and surface characteristics. FLEXPART 11 supports two input formats, data from the European Centre for Medium Range Weather Forecasts (ECMWF) and from the National Center for Atmospheric Predictions (NCEP) forecast systems: Integrated Forecasting System (IFS) and Global Forecast System (GFS), respectively. See section A1 for further details.

For the examples provided in this paper, we use the most recent re-analysis dataset of ECMWF, ERA5 (Hersbach et al., 2020), with hourly $0.5° \times 0.5°$ data as input to FLEXPART. All units are in International System of Units (SI) units, unless otherwise specified.

## 3    Grid-scale advection based on ECMWF fields on native coordinate surfaces

While we know that FLEXPART does not quickly produce strong anomalies in particle distributions in domain-filling simula-
tions (Stohl and James, 2004), the degree to which older versions of FLEXPART obeyed the well-mixed criterion with respect to particle positions over periods of months to years was not entirely satisfying. For this reason, we decided to switch from $z$ to $\eta$ coordinates (see section 3.1), as it might reduce interpolation errors (see section 3.2) and improve particle transport accuracy and particle distribution in domain-filling simulations (see section 3.3).

### 3.1    The $\eta$ coordinate system

The ECMWF's IFS employs a hybrid pressure-base vertical coordinates, where the $\eta$ surfaces are given by

$$\eta_k = a_k/P_0 + b_k \,. \tag{1}$$

Here, $\eta_k$ is the value of $\eta$ at the $k^{th}$ model level, $P_0 = 101325 \,\mathrm{Pa}$ is the reference pressure, and $a_k$ and $b_k$ are coefficients chosen such that the levels closest to the surface follow the topography, the highest levels are pressure surfaces, and intermediate levels are hybrids between the two. The pressure in the $\eta$ coordinate system is determined by the surface pressure $P_s$, which varies



in space and time:

$$P_k(\lambda, \phi, t) = a_k + b_k P_s(\lambda, \phi, t). \tag{2}$$

Here, $\lambda$, $\phi$ and $t$ denote longitude, latitude, and time, respectively. In this system, the vertical velocity is also represented in $\eta$ coordinates, where negative values indicate upward motion. Detailed information on the $\eta$ coordinate system can be found in the ECMWF - IFS documentation (2023). While ECMWF meteorological data can also be downloaded on pressure levels,
pressure-level data are less accurate since they are interpolated between the $\eta$ levels, and fewer levels are available.

The boundary-layer turbulence scheme utilised by FLEXPART (Hanna, 1982; Stohl et al., 1998a) is based on terrain-following geometric height $z$ as a vertical coordinate, with turbulent velocities expressed in units of $\mathrm{ms}^{-1}$. For this reason, and to avoid frequent time-consuming coordinate transformations, older versions of FLEXPART used an internal terrain-following Cartesian ($z$) coordinate system, and all meteorological data were interpolated to these internal $z$ coordinates. However, this
approach introduced additional interpolation errors since the meteorological data needed to be interpolated first from $\eta$ to $z$ levels and then to particle positions.

To mitigate these interpolation errors and improve computational accuracy, FLEXPART 11 retains the $\eta$ coordinate system whenever possible. Only in the ABL, where the Hanna turbulence scheme (Hanna, 1982) is used, the $z$ levels are kept as the main coordinate system. However, interpolation errors are smaller in the ABL than in higher layers of the atmosphere, since
the lowest $\eta$ levels follow the topography and $z$ levels are chosen to coincide closely to $\eta$ levels, with best agreement near the surface and almost perfect agreement for surface pressure values that are typical for sea level. Everywhere else, the native $\eta$ coordinate system is used in FLEXPART 11 to interpolate meteorological data to the particle positions and advect the particles. The $\eta$ coordinates are utilised in FLEXPART 11 by default, but can be switched off at the time of compilation by building the FLEXPART 11 executable with the addition of `eta=no`. In this case, meteorological data are interpolated to $z$ coordinates as
in previous FLEXPART versions. Below, we demonstrate that the switch of coordinate system clearly increases the accuracy of the model computations. We evaluate conservation errors in semi-conserved properties in section 3.2, and the error in the particle density distribution of a global domain-filling simulation in section 3.3.

### 3.2 Semi-conserved properties

It is not a trivial task to prove that changes in a Lagrangian model lead to improved trajectory accuracy, since ground truth
trajectories are usually not available. To show the increase in accuracy obtained as a consequence of keeping ECMWF's $\eta$ coordinate system largely intact (except for within the ABL), we followed Stohl and Seibert (1998) and evaluate the conservation of semi-conserved properties along trajectories (figure 1), such as potential vorticity (PV), potential temperature ($\Theta$), and specific humidity (q). These properties are expected to be reasonably well conserved in the absence of diabatic processes. Therefore, for a date in each season, we released six million particles globally distributed using the domain-filling option, and followed
their trajectories for 48 hours. At heights above 5 km, diabatic processes related to surface interactions can be excluded, and furthermore the differences between $\eta$ and $z$ levels are substantial. Therefore, we traced the trajectories of all particles between 5 and 10, and between 10 and 20 km above ground level using FLEXPART in two simulations using $\eta$ and $z$ coordinates,



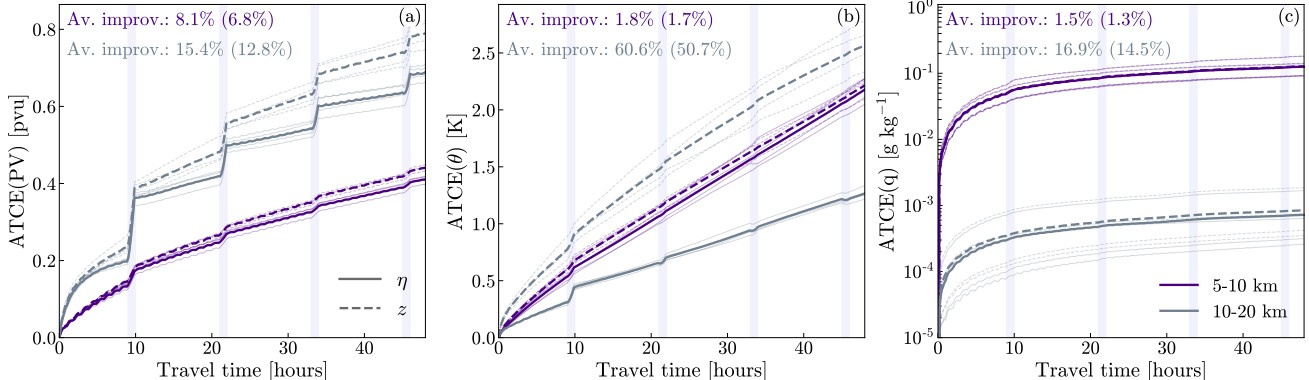

**Figure 1.** Mean absolute tracer conservation errors ($ATCE$) in semi-conserved particle properties: **a)** potential vorticity; **b)** potential temperature; **c)** specific humidity. Results are shown for altitudes between 5 and 10 km (indigo lines) and between 10 and 20 km (grey lines) above ground level as a function of time since the initiation of a trajectory. The figure illustrates these errors for both the $\eta$ (solid lines) and the $z$ vertical coordinate system (dashed lines). Each line represents the mean over around 330000 trajectories that are selected between absolute latitudes of 40 and 80° and a relative humidity below 90 %. Thin lines represent one simulation in each season of 2020, starting on the first day of January, April, July and October at 00 h. Each thick line shows the average of the four seasons. All FLEXPART parameterisations (e. g., turbulence, convection) are switched off. Transition periods between ERA5 12-hourly data assimilation windows are highlighted with lavender shades, beginning 9 hours after the FLEXPART simulation's start. On the top left of each panel, the error reductions that are gained using the $\eta$ coordinates as compared to the $z$ coordinates are given in percentages, both excluding and including (in brackets) data assimilation window transition periods.

respectively. We removed particles that left the specified height range, and to avoid diabatic heating by water phase changes, we removed particles that went through regions where the relative humidity was above 90 %. In addition, to avoid regions

with low Coriolis force, we only used particles at latitudes north of 40°N and south of -40°S. To avoid sampling biases, we selected particles from regions that keep similar density distributions over time (see figure 2), resulting in not including particles that reside north of 80°N or south of -80°S. We also switched off both the turbulence and convection parameterisations in FLEXPART and used short timesteps (`LSYNCTIME=600`).

Using these trajectories, we computed the absolute transport conservation errors (ATCEs) as a function of time along the

trajectory, given by:

$$ATCE(\gamma, t) = \frac{1}{N} \sum_{i=1}^{N} |\gamma_i(t) - \gamma_i(t_0)|, \tag{3}$$

where $\gamma_i(t)$ and $\gamma_i(t_0)$ are the semi-conserved property of particle $i$ at time $t$ and $t_0$ of its trajectory, respectively, and $N$ is the total number of particles in the sample. $ATCE$ values indicate how strongly the considered variables, on average, deviate from perfect conservation along the trajectories.





Figure 1 shows $ATCE$ values for PV (left panel), $\Theta$ (middle panel) and q (right panel) as a function of time along the trajectories. The first thing to notice is that the ATCE values for PV and $\Theta$ show stepwise jumps every 12 hours. These jumps occur during the 1-hour transition of hourly re-analysis data from one 12-hour long data assimilation window to the next in the ERA5 reanalysis production and can be explained by the dynamical inconsistencies between two different assimilation periods. The resulting step-wise increases in conservation errors are a problem inherent to the ERA5 data and not related to

FLEXPART. While the inconsistencies themselves do not necessarily lead directly to trajectory position errors, they do indicate uncertainties in the analyzed state of the atmosphere, which will lead to trajectory position errors. Notably, figure 1 also shows wiggly lines with an hourly periodicity, best visible in the left panel, indicating that the errors grow most rapidly in the middle of two consecutive hourly wind fields. These wiggles can be attributed to the fact that the temporal interpolation errors are largest when the time difference to the closest available wind field is largest (at 30 minutes in between two wind fields).

Most interestingly for our purpose, figure 1 shows that $ATCE$ values for PV, $\Theta$ and q for trajectories between 10 and 20 $\mathrm{km}$ are, respectively, 15.4 %, 60.6 %, and 16.9 % lower for the calculations using $\eta$ coordinates than for those using $z$ coordinates after 48 hours ignoring assimilation window transition periods. Corresponding differences for particles between 5 and 10 $\mathrm{km}$ are 8.1 %, 1.8 %, and 1.5 %, respectively. The relative differences are larger for the higher trajectory starting points, which is expected because our screening to avoid regions of turbulence, convection and clouds is not perfect, and it is to be expected

that the trajectories starting from the lower levels are more strongly affected by diabatic processes there, which can lead to large conservation errors, regardless of which coordination system is used, thereby reducing relative differences between the two trajectory calculations. In summary, our results indicate a substantial reduction in tracer conservation errors when avoiding the interpolation of meteorological data from the original $\eta$ levels to a secondary $z$ coordinate system, and thus an improved accuracy of trajectory calculations with FLEXPART 11 compared to its predecessor versions. While it is impossible to directly

quantify improvements in trajectory position accuracy because of a lack of ground truth data, it is expected that better tracer conservation also indicates more accurate trajectories. Similarly, while we cannot use the dynamical tracers for quantifying conservation errors in the lower troposphere, we certainly also expect improvements in trajectory accuracy there.

### 3.3   Density distribution

Validation of FLEXPART based on the conservation of semi-conserved quantities is restricted to certain regions of the atmo-

sphere and does not directly allow to evaluate the accuracy of trajectory positions. Another validation possibility is to test how accurate the well-mixed criterion is fulfilled. The well-mixed criterion states that if a species of passive marked particles without sources or sinks is initially mixed uniformly in position and velocity space in a turbulent flow, it will stay that way (Thomson, 1987). Generally, Lagrangian transport models could be expected to encounter challenges in this regard as evident from works in the broader literature discussing specific conservational issues for (stochastic) particle methods such as Wang

et al. (2015); Meyer and Jenny (2004); Bahlali et al. (2020), making the test especially relevant. Lagrangian particle models, including FLEXPART (Stohl and Thomson, 1999; Cassiani et al., 2015), are usually tested for this criterion for simulations of turbulent dispersion in the ABL on short time scales. However, this criterion should also be fulfilled (and tested) in the atmosphere as a whole and on long time scales, similarly to what was done byCassiani et al. (2016).



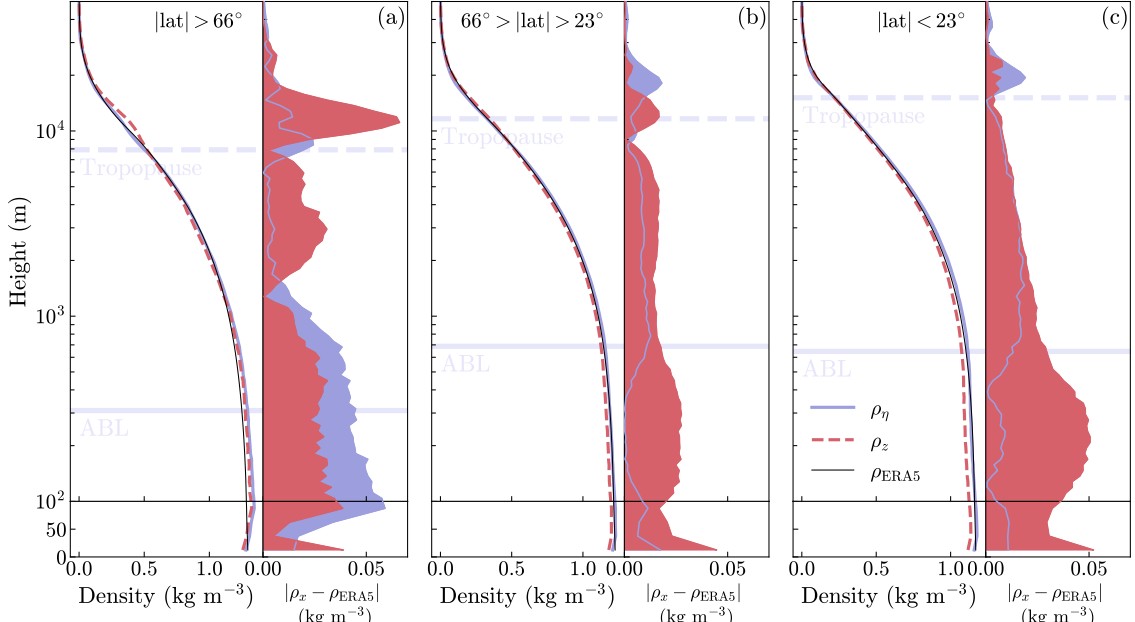

**Figure 2.** Averaged density distribution of particles as compared to the density of air given by the ERA5 data (black lines) after six months of running FLEXPART. The results are averaged over the sixth month (July) of the $\eta$ (blue solid lines) and $z$ (red dashed lines) coordinate system simulations, after initially starting with a perfectly fitting particle density distribution. The results are separated into three regions: Polar regions (**a**), midlatitudes (**b**), and the tropics (**c**). The right hand-side of each panel shows the absolute errors of the $\eta$ (blue) and $z$ (red) simulations. Horizontal thick lavender lines represent the average ABL height (solid) and tropopause height (broken).

To see whether the switch from $z$ to $\eta$ coordinates improved the particle distribution, we performed two six-months global
domain-filling simulations of a passive air tracer based on $z$ and $\eta$ coordinates, respectively. Six million particles were initially
globally distributed, perfectly in accordance with the local air density and, ideally, the particle density distribution should
follow closely the air density distribution throughout the entire simulation, even though the particles were allowed to move
through the atmosphere without any further constraint on their distribution. We used short time steps (`LSYNCTIME=600`, with
both the `CTL` and `IFINE` options set to 10) to increase the accuracy of turbulent transport in the ABL. For evaluation of the
results, we averaged the densities of the particles over the sixth month after the start of the simulations and compared those to
the air densities given by the ECMWF ERA5 data. This is a comprehensive test, as it involves transport in the whole atmosphere
and includes also sub-grid scale parameterisations of convection and turbulence.

Figure 2 shows vertical particle density profiles averaged over polar regions, midlatitudes and tropics, respectively. Overall,
the $\eta$ coordinate system version reproduces the vertical air density profile more accurately than the $z$ coordinate system version.
In the midlatitudes and the tropics, both model runs reproduce the vertical air density profiles relatively well; however, for the
midlatitudes, the $\eta$ version is clearly performing better than the $z$ version throughout almost the entire depth of the troposphere.



For the tropics, better results are seen for the lowest 2 km of the atmosphere. Especially in the ABL, the $z$ version underestimates the air density on average by $\sim 2\,\%$ (midlatitudes) and $\sim 3\,\%$ (tropics). Only in the stratosphere, above about 15 km, the $z$ version performs slightly better than the $\eta$ version. Both model runs overestimate the density below about 1 km in polar regions quite substantially, by up to $\sim 3.6\,\%$. The $z$ version shows somewhat better agreement there, with a weighted average of $\sim 1.2\,\%$ better up to $\sim 1$ km. However, this reverses above about 1 km, and in the polar lower stratosphere, the $z$ version shows a substantial deviation from air density, peaking at $\sim 20.3\,\%$, and here the $\eta$ version shows much better agreement. The underestimation of particle density in the tropical troposphere and overestimation in the polar stratosphere by the $z$ version shows that this version tends to move too many particles from the low to the high latitudes and also to higher altitudes near the poles. This deficiency is less pronounced for the $\eta$ version, which overall shows a much better agreement of particle density profiles with air density profiles in most of the atmosphere (with the exception of the lowest kilometer in the Polar regions and in the stratosphere at lower latitudes). This again indicates the improved trajectory accuracy of FLEXPART 11 compared to previous FLEXPART versions. However, it is also clear that even FLEXPART 11 deviates somewhat from well-mixedness.

We attribute these deviations to a number of possible causes: Firstly, the ECMWF input data are not perfectly mass-consistent, with additional implicit mass-flow violations likely being introduced by the interpolation procedures to generate the sub-grid scale velocity field following the considerations in Wang et al. (2015) and Meyer and Jenny (2004). Secondly, aspects related to grid-scale stochastics could also play a role, as particles may be transferred from regions with higher variability in grid-scale winds towards regions with lower variability because there is no drift correction applied to the grid-scale winds. Lastly, dynamical inconsistencies between data assimilation periods may lead to the production of spurious trajectories.

## 4  Gravitational settling

The gravitational settling of aerosol particles is called at each time step and added to the vertical motion. In FLEXPART 11, a settling module, `settling_mod.f90`, has been introduced containing all the relevant procedures. The module contains several improvements for the calculation of the drag coefficient, $C_d$, of aerosol particles, which is needed to determine their gravitational settling velocity. While previous versions of FLEXPART only considered spherical particles, FLEXPART 11 is able to calculate the settling velocity also for non-spherical particles and takes into account different types of particle orientations (see below). One limitation is that gravitational settling is only calculated if a particle carries a single aerosol species with non-zero mass. This is necessary since aerosols of different size, shape or density exhibit different settling velocities and follow different trajectories. In contrast, a particle can represent any number of gaseous tracers, as they all follow the same trajectory.



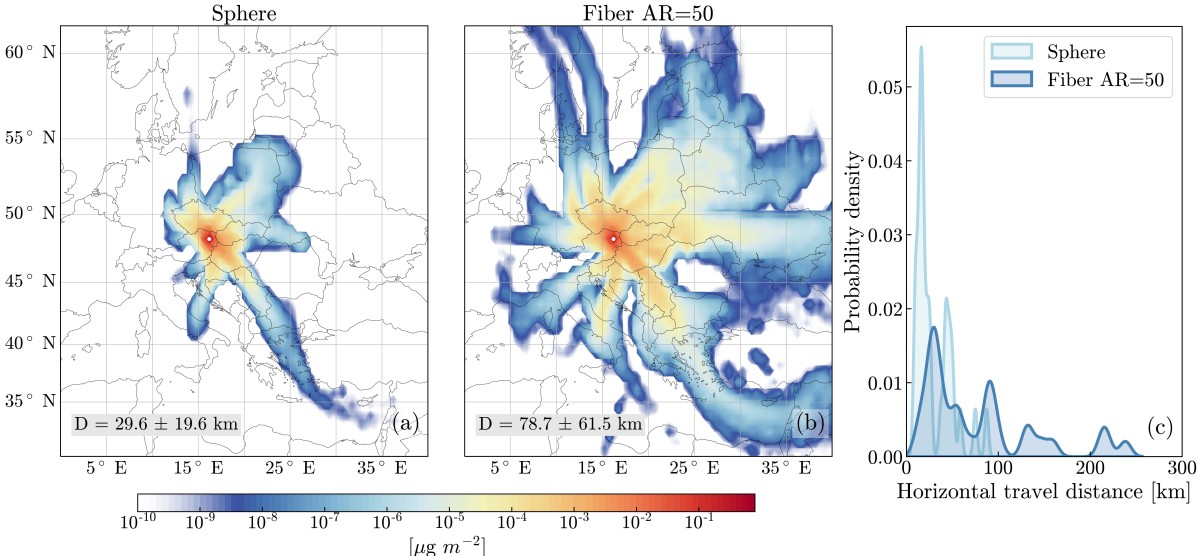

**Figure 3.** Comparison of the atmospheric transport of particles of different shapes. Shown are the FLEXPART simulation results of the monthly averaged total atmospheric deposition for daily releases from a point in Vienna, Austria (white dot), in January 2018 of spheres with a diameter of 50 μm (left panel) and volume-equivalent straight fibres with an aspect ratio (AR) of 50 (middle panel). Values for the monthly mean horizontal transport distance (D) and its standard deviations are also reported near the top in the left and middle panel. The probability density function (PDF) of horizontal travel distance of spheres (light blue) and fibres (dark blue) is displayed in the right panel.

## 4.1 Spherical particles

Earlier versions of FLEXPART only considered the settling of spherical particles. The drag coefficient of spheres in the laminar flow regime can be calculated with Stokes' law. However, Stokes' law is only valid for the case of

$$\text{Re} = \frac{\rho \, d_{\text{eq}} \, w_s}{\mu} \ll 1, \tag{4}$$

where Re is the Reynolds number, $\rho$ the density of air, $d_{eq}$ the volume-equivalent diameter of the particle, $w_s$ the settling velocity and $\mu$ the dynamic viscosity of the air. Substantial deviations from Stokes' law can be expected for particles with diameters larger than 10-30 μm in a laminar regime (Drakaki et al., 2022; Saxby et al., 2018; Alfano et al., 2011). To account also for larger Reynolds numbers, earlier FLEXPART versions used the drag coefficient approximation by Bird et al. (1960). However, this approximation still shows a significant settling velocity mismatch for Reynolds numbers > 400 as compared to other drag coefficient models (Näslund and Thaning, 1991). Therefore, in FLEXPART 11 we replaced this scheme with the drag coefficient formulation of Clift and Gauvin (1971), which is valid for Reynolds numbers up to $10^5$, and is within 6 % of experimentally determined values (Clift et al., 2005):

$$C_d = \frac{24}{\text{Re}} \left(1 + 0.15 \, \text{Re}^{0.687}\right) + \frac{0.42}{1 + \frac{42500}{\text{Re}^{1.16}}}. \tag{5}$$




Using this more accurate scheme changes the settling velocities for large particles substantially. For example, particles with a diameter of 100 μm have a settling velocity lower by about 16 % as compared to the old scheme.

## 4.2 Non-spherical particles

Experiments show that non-spherical particles experience a larger drag in the atmosphere and therefore have lower settling velocities than spheres (Tatsii et al., 2024). To take this into account, we have extended the gravitational settling scheme and the calculation of dry deposition velocities to allow for non-spherical particle shapes. This was done by implementing the scheme of Bagheri and Bonadonna (2016) with modifications by Tatsii et al. (2024). The scheme determines a particle's drag coefficient $C_d$ based on its shape providing three options for the orientation of a particle in the atmosphere: (i) random orientation, (ii) the particle's maximum projection area being perpendicular to the vector of gravity, which corresponds to its maximum-drag or horizontal orientation and thus lowest settling velocity, or (iii) the particle's orientation corresponding to the average of the first two options. The equations defining Stokes' and Newton's drag corrections ($k_S$ and $k_N$) for the three options are given in Table A2. The drag coefficient is then computed as follows:

$$C_d = 24 \frac{k_S}{\text{Re}} \left( 1 + 0.125 \left( \frac{\text{Re}\, k_N}{k_S} \right)^{2/3} \right) + 0.46 \frac{k_N}{1 + 5330 \frac{k_S}{\text{Re}\, k_N}}. \tag{6}$$

The scheme was tested in laboratory experiments with microplastic fibres, where it was found to give best results for the average orientation option (Tatsii et al., 2024). The settling velocities of fibres can be less than one third of those of spherical particles with the same volume, which has major implications for their atmospheric transport and deposition. Figure 3 compares the atmospheric deposition of spheres and equivalent-volume fibres from a point-source release, showing the much greater distances travelled by the fibres.

In terms of technical implementation, parameters defining a particle's shape and orientation are to be provided in the `SPECIES` file. If the three characteristic dimensions (longest axis `PLA`, intermediate axis `PIA`, smallest axis `PSA`) of a non-spherical particle (Blott and Pye, 2008) are known, the user should set `PSHP=1` and provide the three dimension parameters in units of micrometres. If the particle dimensions are not fully known, FLEXPART can compute `PLA`, `PIA` and `PSA` values for a few specific non-spherical particle shapes, assuming that the particles have the same equivalent volume as a sphere with the given diameter `PDIA`. For cylinders (`PSHP=2`), the user also needs to set the aspect ratio `PASPR`, i.e. ratio of the cylinder's length (`PLA`) to its diameter (`PSA`). Other options are cubes (`PSHP=3`), regular tetrahedrons (`PSHP=4`), regular octahedrons (`PSHP=5`), and a regular rotational ellipsoid (`PSHP=6`) characterised by `PLA=2 PIA=2 PSA`. `PSA` needs to be specified by the user. These options are particularly useful for a direct comparison of the atmospheric transport properties of non-spherical particles with those of corresponding spherical particles with the same volume.

The particle orientation in the atmosphere can be specified with the option `PORIENT`: horizontal or maximum-drag (`PORIENT=0`), random orientation (`PORIENT=1`), and the average of the two (`PORIENT=2`). If straight or bent cylindrical particles with aspect ratios equal or greater than 20 are used, it is recommended to select the averaged orientation option as this model best fits the experimental data on gravitational settling of cylinders (Tatsii et al., 2024).



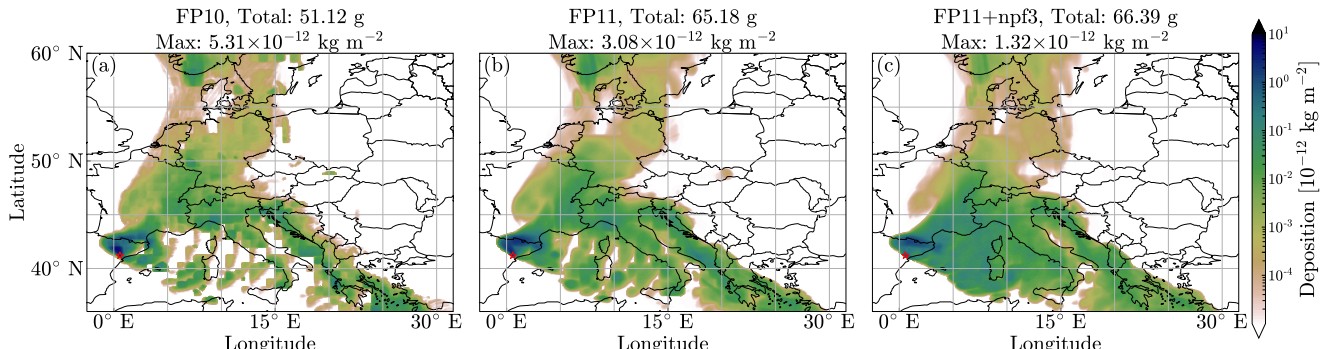

**Figure 4.** Accumulated wet deposition (shading) 66 hours after a point release of 1 kg of aerosols with a diameter of 0.4 μm and a density of $2 \times 10^3 \, \mathrm{kg \, m^{-3}}$ for FLEXPART 10.4 (**a**), FLEXPART 11 with standard precipitation fields (**b**), and FLEXPART 11 using two additional precipitation fields (**c**). 10 million particles were released at constant rate between 18 January 1995 03 UTC and 19 January 1995 3 UTC from the location of the Ascó Nuclear Power Plant (41° 12' N, 0° 3' E, red star). Total and maximum deposition amounts are reported on top of each panel.

## 5   Wet scavenging

Wet scavenging is the process by which trace substances are removed from the atmosphere through precipitation. The mass of particles is reduced and that amount is deposited to the ground. Three main changes were implemented to improve the wet deposition scheme in FLEXPART 11. A full discussion with comprehensive tests will be provided in a forthcoming publication (Tipka et al., in prep.).

### 5.1   New below-cloud scavenging scheme for aerosols

A major change with respect to wet scavenging was introduced in FLEXPART version 8, differentiating between in-cloud and below-cloud scavenging. For the below-cloud scavenging of aerosol particles, the scheme of Laakso et al. (2003) was used with parameters for snow as given by Kyrö et al. (2009). This scheme was derived for small particles only, with diameters between 10 and 510 nm. For FLEXPART 11, a new parameterisation scheme has been introduced that is valid for a wider size range, namely the scheme of Wang et al. (2014), which covers particles from 1 nm (nucleation mode) to 100 μm (coarse mode). This is important especially for larger particles, like dust, sea salt or pollen.

### 5.2   Improved interpolation scheme for precipitation

Version 10 of FLEXPART used a nearest-neighbour method to obtain precipitation rate and cloud information at the location of each particle at the given time; however, everywhere else in FLEXPART, linear interpolation is used. The nearest neighbour scheme for precipitation results in unrealistic 'checkerboard'-like deposition fields, which become visible when time intervals between meteorological fields are large, e. g., 3 h (Hittmeir et al., 2018), and especially in incremental deposition fields. The




underlying problem, which lead the authors of FLEXPART 10.4 to remove linear interpolation of precipitation as implemented before, is related to the fact that the precipitation data is not given as instantaneous rate but as accumulation over a time interval (Hittmeir et al., 2018; Tipka et al., 2020). Hittmeir et al. (2018) therefore proposed a new reconstruction algorithm

for Lagrangian particle dispersion models like FLEXPART which can be applied in conjunction with linear interpolation and which ensures the conservation of integral precipitation within each time interval, maintains continuity at interval boundaries, and conserves non-negativity.

In order to use this new interpolation scheme, the flag `numpf` in `par_mod.f90` must be set to 3. This causes FLEXPART to read in three precipitation fields per input time interval instead of one; these additional fields can be produced with flex_extract

version 7 (Tipka et al., 2020). If this is not set, precipitation will be linearly interpolated between two precipitation fields assigned to the same times as the other quantities, which is still considered an improvement to the nearest neighbour scheme used in FLEXPART 10.4. The example shown in figure 4 clearly illustrates the reduction in artifacts moving from the nearest-neighbour method to simple linear interpolation, and finally the algorithm by Hittmeir et al. (2018) (`numpf=3`). In this example, 3-hourly input data were used, which makes the artifacts more pronounced than with 1-hourly data.

## 5.3  New cloud layer identification scheme

The cloud identification scheme has been improved in FLEXPART 11. One problem is that the cloud water content is provided as instantaneous fields in the ECMWF meteorological input data, whereas precipitation is an accumulated forecast product. Their combination in the scavenging scheme can, at times, produce inconsistencies such as precipitation occurring when no (or only a very thin) cloud is present, and this can lead to unrealistic in-cloud scavenging rates in FLEXPART. To prevent

such unrealistic combinations, the following steps are taken to identify clouds: If cloud water content is available from the meteorological data, the cloud water mass is integrated vertically and used later on for the in-cloud scavenging as already done in version 10. Iterating from the surface upwards, the first instance of cloud water is taken as the height of the bottom of the cloud and the last instance as the top of the cloud, still as in version 10. If, however, the cloud water content is not available, the presence of a cloud is assumed between the lowest level with relative humidity greater than `rhmin`, and the highest level

with that value. Relative humidity is calculated over water for temperatures above -20 ° C, and over ice for temperatures below. The value of `rhmin` is set in `par_mod.f90` (default is 0.90). A minimum thickness of `min_cloudthck` (defined in `par_mod.f90`, default 50 m) is required for a cloud. Spurious clouds below this thickness are eliminated, as they would not be expected to produce precipitation, but if precipitation were present in the data, it would cause unrealistic in-cloud scavenging rates.

For the case of convective precipitation, a simple fix was implemented; it requires that clouds associated with convective precipitation stronger than `precmin` (in `par_mod.f90`; default is 0.002 mm/h) must have a top of at least 6 km and a bottom below 3 km (adjustable by `conv_clrange` in `par_mod.f90`). If this is not fulfilled, the cloud bottom and top will be set to a bottom and top of 0.5 and 8 km for precipitation rates below 0.1 mm/h, and to 0 and 10 km for stronger precipitation (parameters are set in `par_mod.f90`, as `highconvp_clrange`).



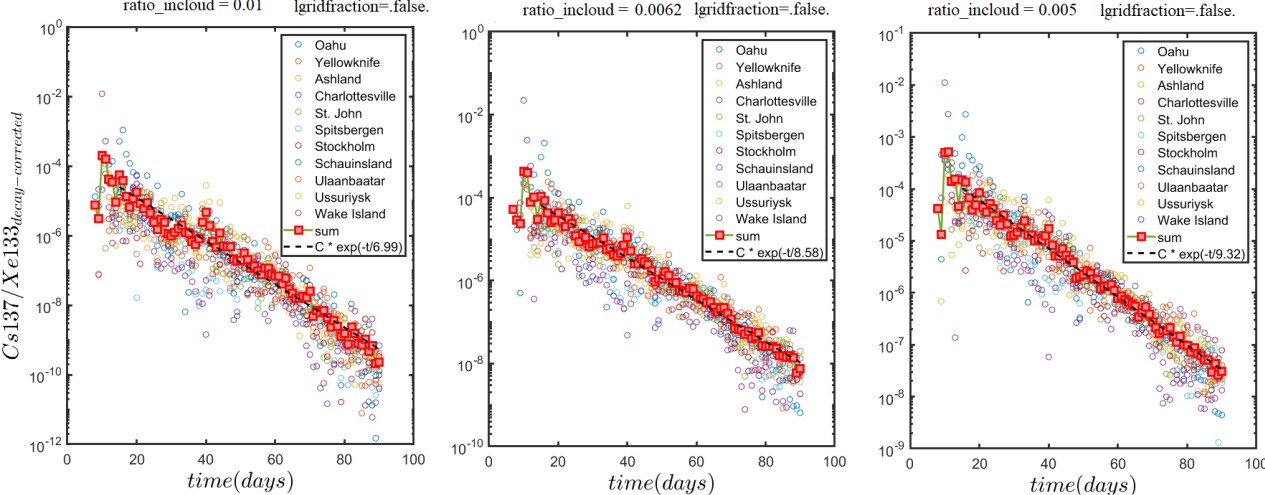

**Figure 5.** Global decrease of the ratio of cesium-137 with respect to xenon-133 (decay corrected) as obtained at several measurement stations as a function of time after the Fukushima nuclear reactor accident. Panels left to right show the results using values of 0.01, 0.0062 and 0.005, respectively, for the replenishment rate $r_{icl}$ of cloud water during precipitation (parameter `ratio_incloud` in FLEXPART 11). Corresponding e-folding lifetimes of the aerosol were 6.99, 8.58 and 9.32 days, respectively.

## 5.4 Fukushima test

To validate that changes made to the precipitation scheme did not introduce errors, several tests have been carried out to check the behaviour of the code with respect to the removal processes. Here, we show a series of tests reproducing the Fukushima Dai-Ichi nuclear power plant accident of March 2011 where the aerosol-bound radioactive isotope cesium-137 ($^{137}$Cs) and the noble gas xenon-133 ($^{133}$Xe) were released in large quantities. As demonstrated by Kristiansen et al. (2016), after correcting for radioactive decay, the ratio of cesium-137 and xenon-133 can be used to evaluate the modelled aerosol lifetimes. They obtained aerosol lifetimes for 19 widely used chemical transport models (including FLEPXART 9) and compared the model results with the aerosol lifetime obtained from experimental measurements. Their results showed for the aerosol e-folding lifetime (among all the models) a model mean of 10.7 days and a model median of 9.4 days. FLEXPART 9 resulted in a shorter aerosol lifetime of 5.8 days using the original Hertel et al. (1995) definition of the cloud liquid water content, $c_l$ (see section A3.2). The observations, as discussed in Kristiansen et al. (2016), suggested a rather longer aerosol lifetime of 14.3 days. More recently, Pisso et al. (2019) re-evaluated the aerosol lifetime in FLEPXART 10.4 with the Grythe et al. (2017) improved scheme and obtained an aerosol decay time of 10 days for cesium-137. Figure 5 shows the aerosol lifetime obtained for three different FLEXPART configurations using different values of `ratio_incloud`, the in-cloud replenishment rate $r_{icl}$ (see section A3.2 for a full description). A decay time of 9.3 days is obtained for FLEXPART 11 when using `ratio_incloud`=0.005, very close to the previously obtained model median by Kristiansen et al. (2016), and therefore the default value set in `par_mod.f90`. A value of `ratio_incloud`=0.001, results not shown here, gave a decay time of about 20 days.



## 6 Linear Chemistry Module

The Linear Chemistry Module (LCM) is based on the initial work of Henne et al. (2018) who developed the FLEXPART-CTM model from FLEXPART 8, and which was first described and evaluated in Groot Zwaaftink et al. (2018). This model was an
350 extension of the domain-filling capability of FLEXPART and added the possibility to initialise particles' mixing ratio from pre-defined fields, account for the influence of surface fluxes and simple linear chemistry on the particles' mass, and sample the particle mixing ratios at user-defined receptor locations.

The essence of the FLEXPART-CTM code has been implemented into FLEXPART 11, which can be run in all its usual configurations, but now including the CTM configuration. Note that the CTM configuration is renamed to Linear Chemistry
Module, reflecting the fact that it is not a separate model to FLEXPART, and that only linear chemical reactions are implemented.

LCM uses the possibility in FLEXPART to fill a global domain with particles that are constantly advected throughout the atmosphere. This scheme was introduced by Stohl and James (2004) for air tracers and has been used for studies of heat and water transport in the atmosphere. However, each particle can carry multiple chemical species. Emissions of a species are not
simulated as releases of new particles representing the emitted mass, like it would be done in a standard FLEXPART run, but are taken up by particles passing by close to the emission source (note, emission in this sense can be positive or negative fluxes; e. g., carbon dioxide could be removed from the atmosphere), thereby changing the species' mass carried by these particles. The mass of the particles for each species can also change due to first order chemical reactions. Deposition of particles is at present not considered.

In the LCM, the mixing ratios are output by sampling the particles on a grid (specified by `OUTGRID`), at receptor locations (specified by `RECEPTORS` and, new in FLEXPART 11, for satellite receptors (specified by NetCDF files generated from a satellite pre-processor, `prep_satellite`, and can be created using software obtainable from https://git.nilu.no/flexpart/flexinvertplus). The mixing ratios for each species are calculated using the ratio of the mass of the species over the mass of air, where the mass of air is always carried by species number 1. This method was chosen over using the mass of the species
divided by air density and volume, which is the standard method, because it leads to less noisy results and avoids problems with spurious accumulation of particles (see section 3.3).

The LCM mode is switched on in FLEXPART in `COMMAND` with the switch `LCMMODE=1`. In addition, the following options in `COMMAND` should be set: `MDOMAINFILL=1`, `IND_SOURCE=1`, and `IND_RECEPTOR=1`. When these options are set, mixing ratios will be calculated using the ratio of masses method. The domain-filling mode uses the `RELEASES` file to define
the species, the domain and the total number of particles to use. For LCM, the first species in `RELEASES` must always be `AIRTRACER`.

A full discussion with tests and examples will be provided in a future publication (Thompson et al., in prep.). Nudging to observations, as implemented in FLEXPART CTM (Groot Zwaaftink et al., 2018), is not yet available in the LCM module but will be added in the near future.



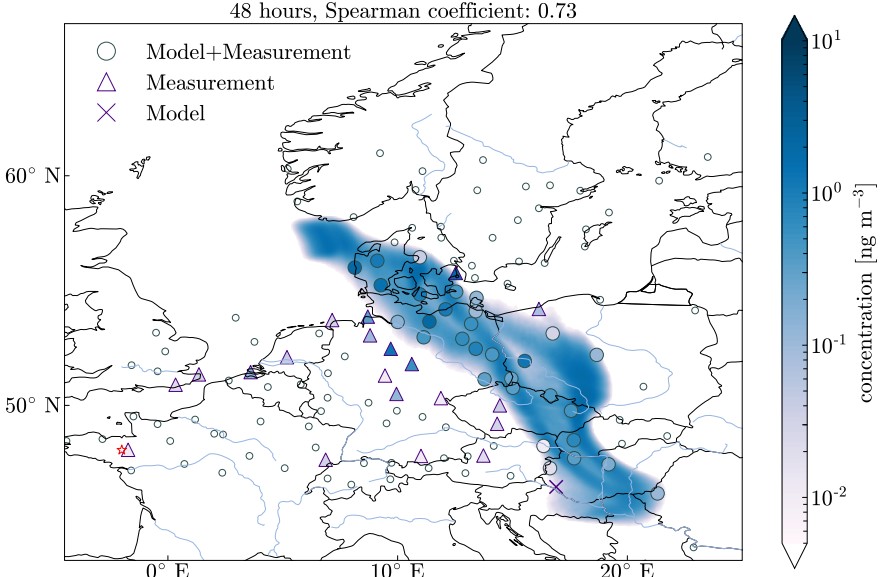

**Figure 6.** Comparison of the concentration fields obtained by FLEXPART 11 (blue contours), using $\eta$ coordinates on a $0.1\,^{\circ} \times 0.1\,^{\circ}$ grid, and the ETEX measurements (markers) $48\,\mathrm{hours}$ after the ETEX-1 release. Small grey circles represent each zero measurement that corresponds to a zero model result. Large grey circles represent the stations with non-zero measurements where the model also found a signal. Indigo triangles represent stations with non-zero measurement where the model found no significant signal, and indigo crosses show where the opposite is the case. Shades within the circles and triangles show the concentrations measured at the corresponding stations. The initial release location near Rennes is marked by a red star.

## 7  Comparison with tracer experiments

In section 3, we have shown how semi-conserved properties of particles are better conserved and well-mixedness better fulfilled when using $\eta$ instead of $z$ vertical coordinates. However, we made large modifications also in other parts of the source code, and therefore it was important to validate FLEXPART 11 for a large range of different applications, including output of gridded properties. One excellent validation possibility is to compare the model against tracer experiments. We chose to follow Stohl et al. (1998b), and applied FLEXPART 11 to the first European tracer experiment (ETEX) (Nodop et al., 1998) and the Cross-Appalachian tracer experiment (CAPTEX) (Raynor et al., 1984; Ferber et al., 1986). Following Stohl et al. (1998b), we compute:

– the Normalised Mean Square Error: $\mathrm{NMSE} = (1/N) \sum_{i=1}^{N} (P_i - M_i)^2 / (\overline{P} \cdot \overline{M})$, where $P$ and $M$ represent the model predictions and measurements, respectively, and $N$ is the number of measurements,

– the Root Mean Square Error: $\mathrm{RMSE} = \sqrt{(1/N) \sum_{i=1}^{N} (P_i - M_i)^2}$, where $P$ and $M$ represent the model predictions and measurements, respectively, and $N$ is the number of measurements,



**Table 1.** Statistical measures of how well FLEXPART 10.4 and FLEXPART 11 using $\eta$ and $z$ coordinate systems, respectively perform as compared to the ETEX observations (top) and as compared to the CAPTEX aircraft measurements (bottom). CAPTEX results show the mean statistics over six releases. Number gives the total number of model-observation pairs used for the analysis. The RMSE is given in $\mathrm{ng\,m^{-3}}$.

| ETEX | $n$ | NMSE | RMSE | FMS | SCC | PCC | FA2 | FA5 | FOEX |
|---|---|---|---|---|---|---|---|---|---|
| Fp10 | 3104 | 1607 | 5.90 | 47.9 | 0.674 | 0.630 | 0.92 | 0.95 | -19.6 |
| Fp11 $\eta$-coord. | 3104 | 1540 | 5.63 | 47.1 | 0.672 | 0.630 | 0.92 | 0.96 | -20.7 |
| Fp11 $z$-coord. | 3104 | 1546 | 5.63 | 47.2 | 0.674 | 0.630 | 0.92 | 0.96 | -21.6 |
| **CAPTEX** | | | | | | | | | |
| Fp 10 | 203 | 91.47 | 8.90 | 47.64 | 0.535 | 0.332 | 0.677 | 0.745 | -4.81 |
| Fp11 $\eta$-coord. | 203 | 78.47 | 9.13 | 47.93 | 0.559 | 0.331 | 0.652 | 0.717 | -0.16 |
| Fp11 $z$-coord. | 203 | 78.39 | 9.11 | 47.73 | 0.565 | 0.332 | 0.652 | 0.718 | -0.26 |

- the Spearman rank-ordered (SCC) correlation coefficient,

- the Pearson (PCC) correlation coefficient,

- the Figure of Merit in Space: $\mathrm{FMS} = 100 A_\mathrm{p} \cap A_\mathrm{m} / A_\mathrm{p} \cup A_\mathrm{m}$, where $A_\mathrm{p}$ and $A_\mathrm{m}$ represent the model predicted and measured concentrations above $0.1\ \mathrm{ng\,m^{-3}}$, respectively,

- the fraction of model predictions that lie within 2 times (FA2) and 5 times (FA5) the measured values,

- the frequency of over- and under predictions: $\mathrm{FOEX} = 100\left(N_{P_i > M_i}/N - 0.5\right)$, with $N_{P_i > M_i}$ the number of overpredictions.

We computed the statistical measures for two different settings of FLEXPART 11, using the $\eta$ coordinates and $z$ coordinates. To ensure consistency with the previous FLEXPART version, we also calculated the statistics for FLEXPART 10.4, using the same input options as for FLEXPART 11. We used ERA5 meteorological input data, simulated a passive tracer and used time steps shorter than 10 % of the Lagrangian turbulent time scale (settings `CTL=10` and `IFINE=10`). We use an output grid with $0.1\,° \times 0.1\,°$ resolution.

## 7.1 ETEX

On 23 October 1994, 340 $\mathrm{kg}$ perfluoromethylcyclohexane (PMCH) were emitted near Rennes, France. The release lasted 12 hours, and measurements were taken for the next 90 hours across 168 stations distributed all over Europe (see figure 6). We recreated this experiment by releasing 1 million particles equally spread over the 12 hours time frame with altitudes between 8 and 20 meters. The results of the statistical measures are shown in table 1. The versions using the $z$ and $\eta$ coordinate systems give almost identical results. This is not unexpected, as most of the gas stays in the turbulent ABL where the $z$ coordinate system



is always used. Importantly, FLEXPART 11 using the $z$ coordinate system version produced similar results as FLEXPART 10.4, which demonstrates that the many code changes did not result in undesired performance losses. In fact, with the exception of FA5 and FOEX, all statistical values are slightly better for FLEXPART 11 than for FLEXPART 10.

## 7.2 CAPTEX

During the Cross-Appalachian tracer experiment (CAPTEX), seven separate PMCH releases were made in North-America
(Ferber et al., 1986) and both ground-based and airborne PMCH concentration measurements were made. Even though all aircraft measurements were made below 3053 m above sea level, and many within the ABL, the values at these altitudes are more likely to be affected by the changes in the coordinate system than those at the ground; therefore, we present FLEXPART results only for the aircraft measurements. To compare the difference in densities between the aircraft measurements and the FLEXPART output, we use the mass of particles present in a volume of a 0.5° by 0.5° by 100 meters box around the
measurement locations.

In table 1 we list the average and medians of the statistical measures for all CAPTEX releases, excluding release 6, for which the number of valid observations was very small. We show the mean of the statistical indicators, instead of highlighting the spread in accuracy between the different releases. Table 1 mainly shows that using the $\eta$ coordinate system as compared to the $z$ coordinates version leads to no substantial differences, although the FOEX values indicate that the $\eta$ coordinates version
shows less bias compared to the measurements. We also see no systematic large differences between FLEXPART 10.4 and FLEXPART 11, except for the NMSE values which again are better for FLEXPART 11.

## 8 OpenMP parallelisation

Pisso et al. (2019) implemented a pure Message Passing Interface (MPI) method for the parallelisation of FLEXPART. MPI often performs equally well or better than other multi-processing options without excessive communication between processes,
and potentially low overhead. However, FLEXPART runs on input meteorological data that is ever growing in size, and without domain decomposition, each MPI process needs to keep a copy of the same meteorological data in memory. This leads to a large memory footprint that can easily exhaust the memory of a compute node. Consequently, we have followed a different parallelisation strategy for FLEXPART 11 that is based on Open Multi-Processing (OpenMP), which shares memory between processes. While the OpenMP option limits the number of parallel processes to the number of cores available on a compute
node, this is not a significant limitation for typical FLEXPART applications. Furthermore, because of the linear nature of FLEXPART calculations, trivial parameterisation by splitting a simulation into several separate run instances, each running on a different node, is always possible for particularly large simulations. One drawback of OpenMP compared to serial code is that it is more difficult to modify it. To avoid errors related to parallel execution, users unfamiliar with OpenMP are strongly advised to make changes in the form of subroutines and functions and avoid the use of global variables.





## 8.1 Parallelisation strategy

The way how the computational time used in a FLEXPART run is partitioned between different parts of the code depends on the set-up and thus on the peculiarities of the application considered. For example, if many particles are released from a single point and only a small computational domain is used, initially most of the time is spent on particle trajectory computations. However, when a global high-resolution domain for the meteorological input data is used in a global domain-filling simulation, a significant share of the CPU time is spent on the convection computations. On the other hand, if relatively few particles are used, computations related to the gridded meteorological input data (e. g., coordinate transformations) are taking a larger share. For this reason, the OpenMP parallelisation was implemented throughout the various components of the model, trying to avoid bottlenecks for all these cases, especially for the most common set-ups.

We parallelised all particle based computations, apart from their initial release in the `releaseparticles` subroutine. On top of that, we parallelised the convection, wet and dry deposition, and the vertical coordinate transformation of the fields. Lastly, we parallelised the computations needed for the output, both for the gridded output and the particle dump.

The total number of threads for gridded-output-related computations can be set by the user (MAXTHREADGRID=<number of threads> in the `COMMAND` file) with a maximum of the general requested number of threads. The reason for this is that parallelisation is applied to the particles, and therefore each thread needs its own set of output grid variables. Depending on the resolution of the output grid, this can significantly increase the memory required. On top of that, depending on the number of particles in the simulation, the related computational overhead might outweigh the speed-up. The default of MAXTHREADGRID is set to a maximum of 1 task, thus switching off parallel computations related to the grid. A number of threads >1 should only be set after having tested with the specific set-up to be used. For example, for a simulation with 10 million particles and a $0.5°$ by $0.5°$ global output grid, using more than 16 threads will degrade performance (with the optimum number being 6), while for a simulation with only 10 thousand particles and a $0.1°$ by $0.1°$ output grid, parallelisation is not at all efficient and MAXTHREADGRID should be left at its default value of 1.

## 8.2 Code performance and scalability

To evaluate the scalability of the OpenMP parallelisation across different FLEXPART running options, we select a few test cases across the spectrum of typical FLEXPART applications:

– **Case Tracer:** Domain-filling simulation with particles representing a passive air tracer distributed across the globe following air density. Every hour, all particle information but no gridded output is written to NetCDF files (see section 9.1). We run FLEXPART for 5 hours, using 10 minute time-steps, and we set the turbulence options CTL=10 and IFINE=10. This option covers a large range of different computations within FLEXPART (i.e. convection and turbulence in all possible conditions, particle output, etc.) and is therefore a difficult condition to obtain perfect scalability for.

– **Case Aerosol:** Aerosol particles that are initially homogeneously distributed in the bottom 100 meters across the globe. Every hour, gridded properties are printed to NetCDF files on the same horizontal resolution as the input data ($0.5°$ by $0.5°$ global grid), and four vertical levels. FLEXPART is run for 5 hours, using 15 minute time-steps, and turbulence

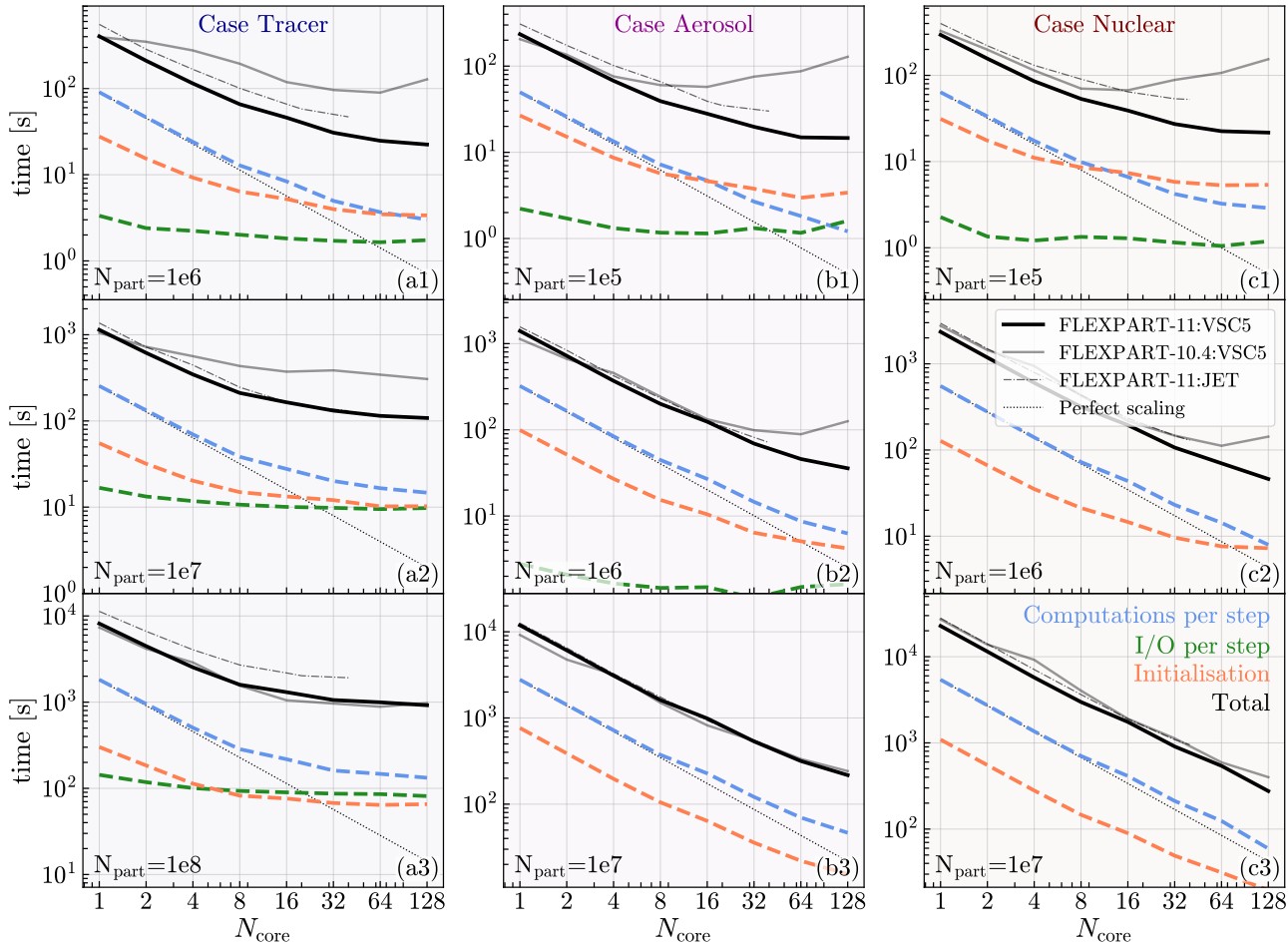

**Figure 7.** Demonstration of the strong scalability of the OpenMP parallelisation of FLEXPART 11. Shown is the computation time as a function of the number of cores used ($N_{core}$). Perfect strong scalability is represented by the slope of the straight dotted thin black lines. The left column of panels, shaded in blue, show results for globally distributed passive tracer particles (**Case Tracer**), using $10^6$ (a1), $10^7$ (a2), and $10^8$ (a3) particles. The middle column, shaded in magenta, shows results for globally distributed aerosols (**Case Aerosol**) using $10^5$ (b1), $10^6$ (b2), and $10^7$ (b3) particles. The right column, shaded in maroon, shows results for xenon-133 emitted from a single point source (**Case Nuclear**) using $10^5$ (c1), $10^6$ (c2), and $10^7$ (c3) particles. In black, the total of 5 hour running time is plotted for FLEXPART 11 (black solid lines). The MPI parallelised FLEXPART 10.4 equivalent is plotted for reference (solid grey lines). The coloured dashed lines represent the break-down of computational time spent in different components of FLEXPART 11: the blue lines represent the total computational time per hourly time-steps minus the I/O operations (green lines), and initialisation of the run (orange lines). Note that for some cases, green lines can not be seen, as the I/O operations fall below the limits of the y-axis, but are of similar order as the I/O operations of the same case with fewer particles. The runs represented by the thick lines and the FLEXPART 10.4 run were executed on the VSC-5 machine, and the thin dashed-dotted black lines show total FLEXPART 11 running time on our local server Jet. Note that the black lines are not the sum of all other lines, since the black lines show the total time the simulation took, while both the I/O operations and computations are shown per hour of simulation time. This was done for clarity of presentation, i.e., to better separate the lines from each other.







**Figure 8.** The memory footprint of FLEXPART 11 (solid thick lines for $\eta$, broken lines for $z$ coordinate system) and FLEXPART 10.4 (dotted thin lines) simulations as a function of number of cores. Shown are the results for Case "Tracer", a domain-filling simulation using 1 million (purple), 10 million (blue), and 100 million (green) particles. Note that the FLEXPART 11 cases using 1 and 10 million particles largely overlap.



**Table 2.** Weak scaling and serial comparison of different applications described in section 8.2. **Weak scaling** is defined by the time it takes to run A (top row) particles on 1 core compared to running B particles on 10 cores: $[t(B_{10}) - t(A_1)]/t(A_1)$. The **serial comparison** is done by comparing the time it takes to run ten times A particles and a single B particle run, both on 1 core: $[t(B_1) - 10t(A_1)]/10t(A_1)$. All values are given in percentages.

| | | A: 100 thousand B: 1 million | | A: 1 million B: 10 million | | A: 10 million B: 100 million | |
|---|---|---|---|---|---|---|---|
| | | VSC | Jet | VSC | Jet | VSC | Jet |
| Tracer | weak scaling | | | -51 | -61 | +33 | +89 |
| | serial comparison | | | -71 | -74 | -28 | -5 |
| Aerosol | weak scaling | -20 | -30 | +5 | 0 | | |
| | serial comparison | -37 | -48 | -15 | -21 | | |
| Nuclear | weak scaling | +1 | 0 | +14 | +8 | | |
| | serial comparison | -16 | -26 | -3 | -5 | | |

options `CTL=10` and `IFINE=4`. Case Aerosol simulations generally take much longer than Case Tracer simulations, on one hand because of the extra computations in the wet and dry deposition and gravitational settling routines and on the other hand because of all particles starting within the ABL, where solving the Langevin equations of the turbulence parameterisation requires very short time steps. We set `MAXTHREADGRID` to 16, meaning that the gridded computations are using a maximum of 16 threads.

   – **Case Nuclear:** A single xenon-133 point release, using the CBL option for skewed turbulence in the ABL and `CTL=40` and `IFINE=5`, with a nested input domain and nested gridded output over Europe with $0.25° \times 0.25°$ resolution. Unlike with MPI parallelisation, OpenMP parallelisation is only active within specified regions of the code. With this third case, we want to complete our demonstration of the parallelisation of all possible parts of FLEXPART, including radioactive decay, skewed turbulence, and computations on the nested grid. Gridded computations are conducted on a single thread.

We use two different computers to demonstrate the performance of FLEXPART 11 across platforms. The first one is the local department server (called "Jet"), which has 9 Intel Skylake compute nodes, each with 2x20 cores and 80 threads in total and 768 GB memory. The second one is the Vienna Scientific Cluster 5 (VSC-5), which has 564 AMD EPYC Milan compute nodes, each with $2 \times 64$ cores and 256 threads in total, 512 GB RAM and 1.92 TB SSD storage. For both computers, we use the GNU Fortran compiler with the optimisation flag `-O3`, and since we have Intel CPUs on Jet, we use `-march=skylake-avx512` for the compiler to correctly recognise the hardware architecture. The compiler versions are GNU compiler collection (GCC) 8.5.0 on Jet, and GCC 12.2.0 on the VSC-5. We set `export OMP_PLACES=cores`, and `export OMP_PROC_BIND=true`, for the best use of non-uniform memory access (NUMA) and therefore fastest performance. To make the comparison between the two versions clearer, we do not make use of the hyperthreading capabilities of the





computers. This means that the number of OpenMP threads are equal to the number of tasks used by MPI and we use threads and cores interchangeably when referring to OpenMP processes.

We investigate both the strong and weak scalability of our OpenMP parallelisation for each case. Strong scaling refers to
the speed-up by increasing the number of threads: Ideally, the simulation time should scale close to the inverse number of cores, $1/N_{\mathrm{cores}}$ for all cases (e. g., ideally giving a speed-up of 10 when using 10 times the number of cores). We separated the initialisation, which only happens once and would thus become relatively less important for longer model runs than our rather short 5-hour test cases, and I/O operations, which are very dependent on the output needs of the user (e. g., output frequency of the particle dump), from the computations that are done every time-step, which probably best represent the
overall model performance in realistic cases. Weak scaling refers to the relative speed when increasing the size of the problem in proportion with increasing the number of cores used for the computation (e. g., running 10 times the problem size on 10 times the computational resources should ideally take the same amount or less time). In our case, we only increase the number of particles but keep the size of the meteorological input data constant for all parallel runs, since most users do not switch between data types. That means that the problem size does not increase entirely tenfold when running ten times the number of
particles.

Scalability of computation times is not the only issue to consider for evaluating the FLEXPART performance. Equally important is the memory requirement of the model. Therefore, we also compare the memory footprint of FLEXPART 11 with the MPI parallelised version of FLEXPART 10.

### 8.2.1 Case Tracer

The left column of figure 7 shows how the strong scalability for Case Tracer depends on the number of particles used in a simulation (solid black lines) by benchmarking FLEXPART 11 using 1 million (top), 10 million (middle), and 100 million particles (bottom). Every panel also shows a comparison to the FLEXPART 10.4 MPI version (dashed black lines). For 100 million particles, FLEXPART 10.4 with MPI and FLEXPART 11 are performing similarly, while for smaller particle numbers FLEXPART 11 clearly scales better than FLEXPART 10.4. The scalability of the main computations (blue lines) are dependent
on the number of particles that are used; for example, on VSC-5, when using 32 cores, we find a $1.8$, $2.5$, $2.8$-fold increase in computational time as compared to perfect scaling for the three problem sizes of 1 million, 10 million, and 100 million particles, respectively. On Jet, the performance is worse, with $2.3$, $3.0$, $4.1$, for the three problem sizes, respectively. The large difference between VSC-5 and Jet is likely the result of using a more recent version of the GCC compiler on VSC-5 compared to Jet (GCC 12.2.0 instead of GCC 8.5.0), which optimises the non-parallelised regions more effectively. The time
it takes for writing the NetCDF particle dump files is less than 10 % of all the computations per step when using only a few cores. This increases to up to 43.1 % when using 128 cores because the writing of NetCDF files cannot be parallelised using OpenMP. Since the output size for the Case Tracer increases linearly with the number of particles used, total scalability suffers from the relatively inefficient writing of output files when using many cores.

The weak scalability depends heavily on the number of particles of the simulation. We find a decrease of more than 50 %
in computation time when running 10 million particles on 10 cores instead of running 1 million particles on 1 core (see the





top row of table 2), which can be explained by parallelisation of the convection computations, since these take an (almost) equal amount of time regardless of the number of particles in a grid column. Increasing the number of particles from 10 to 100 million, however, increases the computation time by more than a third. Profiling indicates the likely cause to be inefficient communication between NUMA regions. Therefore, for large problem sizes, it could pay off to parallelise by splitting the

problem into multiple smaller ones that are simulated separately. However, fewer CPU hours in total will be consumed when simulating more particles at once on a single core: the serial comparison in table 2 shows that in the range of 1-100 million particles, it always takes less time to simulate more particles at once than splitting them up into multiple smaller runs.

### 8.2.2 Case Aerosol

For Case Aerosol (middle column of figure 7), we chose to use 100 thousand, 1 million and 10 million particles for our

benchmark testing, separating the computation in the same way as described in Case Tracer. Generally, the performance of FLEXPART 11 is similar to or better than that of FLEXPART 10.4, with the exception of the serial version, where FLEXPART 10.4 is faster. Large improvements in performance for version 11 are especially visible when using smaller numbers of particles on more cores, where the MPI version suffers from more overhead.

The main computations (without I/O and initialisation) of Case Aerosol generally have better scalability than Case Tracer,

coming close to perfect scaling for all particle number cases, especially when using up to 8 cores. For the example of 32 cores, scaling is only 1.7 (2.5), 1.4 (2.0), and 1.4 (1.9) times slower for 100 thousand, 1 million and 10 million particles on VSC-5 (Jet), respectively, than perfectly scaled code would be. The I/O per step, reading meteorological data and the organisation and writing of information to the grid in NetCDF format, is only slightly improving with the number of cores, but is a significant bottleneck only for the smallest problem size (100 thousand particles). Here, it takes up $\sim 4\,\%$ of the time of all computations

in serial mode and $\sim 57\,\%$ when using all 128 cores, as a consequence of the excellent scaling of the other computations. The largest problem size reports a minimum of $\sim 0.2\,\%$ (serial mode) and a maximum of $\sim 8\,\%$ (128 cores) time being spent on I/O operations.

The weak scalability of the main computations is adequate as shown in the middle row of table 2, taking about a fifth less time to compute a time step of 1 million particles on 10 cores as compared to 100 thousand particles on 1 core, but taking

slightly more time for the larger problem size. When running FLEXPART in serial mode, less CPU hours are consumed by simulating many particles at once as compared to splitting the problem into multiple smaller simulations, although this gain decreases when moving towards larger problem sizes.

### 8.2.3 Case Nuclear

For Case Nuclear (right column of figure 7), we chose a single release of 100 thousand, 1 million and 10 million particles.

Similar to Case Aerosol, the strong scaling of the main computations for 1 million and 10 million particles is reasonably good, only being 1.3 (1.4) and 1.3 (1.2) times slower, respectively, than perfect scaling when using 32 cores on VSC-5 (Jet). For the 100 thousand particles run, this becomes 2.1 (2.9) times slower. For all problem sizes and number of cores, FLEXPART 11 outperforms FLEXPART 10. The I/O computations take no more than $\sim 3.4\,\%$ of all computations per step in serial mode,



and a maximum of $\sim 36\,\%$ when using the full 128 cores for the smallest problem size. For the largest, this ranges between $\sim 0.08\,\%$ (serial) and $\sim 8.2\,\%$ (128 cores).

The weak scaling relation is within the range of the other two cases. Table 2 shows that a run using 1 million particles divided on 10 cores, takes approximately the same time as simulating 100 thousand particles on one core. And a run using 10 million particles divided on 10 cores, takes between $\sim 10-15\,\%$ more time than a run of 1 million particles on only 1 core. However, when running FLEXPART on a single core, less CPU hours are consumed by simulating many particles at once as compared to splitting the problem into multiple smaller simulations, although this gain is marginal when requiring a total number of more than 10 million particles.

### 8.2.4 Memory requirements

In the previous section we showed that in most cases, OpenMP parallelisation results in better speed-up than the MPI parallelisation of FLEXPART 10.4. However, the main advantage of using OpenMP over MPI is that data (especially the meteorological input fields) are shared between the processes, instead of requiring separate copies on each core. That reduces the memory footprint substantially for runs on shared-memory systems. Typically, work group servers and most high-performance computers today provide many cores per node but only a limited amount (some GB) of RAM per core. For example, as shown in figure 8, when running FLEXPART 11 on Case Tracer with $10^6$ particles, approximately 11 GB of memory are used, regardless of the number of cores used. An equivalent simulation with FLEXPART 10.4 on 32 cores, requires about 132 GB, more than ten times the memory footprint of FLEXPART 11. Running FLEXPART 10.4 on one full node with 128 cores on the VSC-5 exceeds the node's available memory of 500 GB. We note that without parallelisation, FLEXPART 11 needs somewhat more memory than FLEXPART 10.4 (e. g., 11 GB instead of 7 GB for the example above), which is mainly due to the extra memory needed for the $\eta$ coordinate system, moving to a necessary double precision default for some variables in FLEXPART 11, and added functionality.

## 9 Running FLEXPART: New input and output options

In this section, we will give an overview of new user functionalities of FLEXPART 11. For general information about how to run FLEXPART, we strongly recommend for users to check the instruction manual (https://flexpart.img.univie.ac.at/docs), since this page will be kept up to date with any future additions and changes.

### 9.1 Particle output

In addition to the gridded output, FLEXPART can also write out information related to each particle, its position as well as meteorological information associated with it. In FLEXPART 10.4, this was possible already, and for reasons of efficiency, unformatted binary files per output time step were produced. Since users found it cumbersome to work with these, for FLEXPART 11, particle information is now written to NetCDF files instead. With HDF5-based compression, NetCDF files are even about $\sim 50-60\,\%$ smaller than their binary equivalents. Another advantage of the NetCDF output is that it is easier to modify





the content or structure of the information to be written, especially with respect to safe post-processing. Taking advantage of this, it is now possible to specify which particle properties are to be written out. The desired parameters can now be specified in a new option file, `PARTOPTIONS`. It is now required if particle properties are to be written out (i.e., when `IPOUT=1` is set in the `COMMAND` file). Currently, the available variables are: particle position (longitude, latitude, height in meters above ground), PV, specific humidity, air density, temperature, pressure, particle mass, cumulative loss of mass by dry and wet deposition

(separately), settling velocity, 3D grid-scale particle velocities, the height of the ABL, the tropopause, and the topography. Each property can be written out as an average over the preceding output time interval, and / or as instantaneous values.

In previous FLEXPART versions, the memory location of terminated particles was reused for newly released particles in order to minimise the memory requirement. This, however, complicates the tracking of individual particles. Hence, in FLEX-PART 11, if particle output is switched on (`IPOUT=1`), terminated particles are kept in the simulation, but values associated

with them are set to 'NaN' instead of being overwritten by newly released particles in the NetCDF output. This comes with no additional computational cost, but it may require more memory than running without the particle output option switched on. The previous behaviour of overwriting terminated particles when using particle output can be restored as explained in the instruction manual (https://flexpart.img.univie.ac.at/docs) and will be needed for domain-filling simulations with a limited domain, where particles are continuously produced at the inflowing boundary and destroyed at the outflowing boundary.

## 9.2 Starting a simulation from user-defined particle data

To give users complete control over the initial conditions of a simulation, in FLEXPART 11 we implemented an option to start a simulation from particle information provided in an additional input file in NetCDF format. This file must hold information about the initial particle positions, their time of release, and their initial mass (see the manual for a full description). This option is used if `IPIN=3` is set (`IPIN=4` in the case of a simulation to be restarted). The user then needs to provide a file

`part_ic.nc` in the output directory as specified in the `pathnames` file. The `RELEASES` file in the input directory is ignored in this case. This option is particularly useful if release geometries are complicated and latitude-longitude-altitude boxes are not suitable. In Bucci et al. (under review), this option was used to produce a global release at the ocean surface. Other possible applications are FLEPXART runs for satellite observations, with non-trivial geometry of the pixel at the Earth's surface and height-varying kernel sensitivities. In such a case, for a backward simulation to determine the sensitivities of column-average

molar fractions to emissions and/or initial conditions, one would fill the volume relevant for a specific satellite measurement with particles, horizontally homogeneous and with a vertical profile proportional to the pressure-interval-weighted satellite retrieval kernel. Figure 9 shows such an example for a satellite observation over India.

## 9.3 Restarting a simulation

As a FLEXPART run may be interrupted (e. g., because of server maintenance, reaching the maximum allowed wall-clock

time, etc.), already in FLEXPART 10.4 there was an option to restart a simulation from a saved simulation state. However, this required the user to request a particle dump to binary files, which was done at every output time step; this could substantially slow down the whole simulation. Furthermore, the gridded deposition data lacked the information from the previous run. In



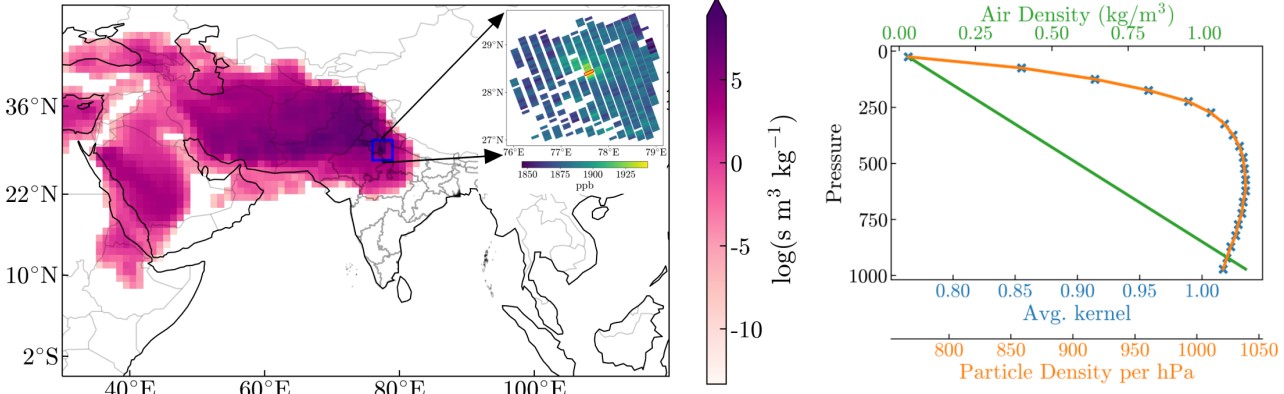

**Figure 9.** Methane emission sensitivity produced with a 7-day FLEXPART backward simulation from a TROPOMI methane remote sensing observation (Schneising et al., 2023). **(a)** Emission sensitivity obtained for the location of the TROPOMI observation. The inset **(c)** shows all TROPOMI methane observations that were made in the region inside the blue square, with one single observation near the centre highlighted by a red outline. The FLEXPART backward simulation was done for this single observation. **(b)** Vertical profile of the FLEXPART particle density per pressure interval (given as number of particles per hPa). The number of particles, $P_n$, released in layer $n$ of the 20 TROPOMI retrieval layers was calculated as $P_n = P \cdot A_n \cdot W_n$, where $P$ is the total number of particles used for the simulation (1 million), $A_n$ is the averaging kernel value for retrieval layer $n$, and $W_n$ is the pressure weight. The particles are uniformly distributed horizontally based on the geometry of the satellite data pixel (i.e., the pixel marked with a red outline in panel a). Also shown in panel b is the averaging kernel and the air density. FLEXPART particle releases were made using the user-defined initial particle conditions option (file `part_ic.nc`).

FLEXPART 11, we created a separate option in the `COMMAND` file which sets the time interval for the output of the binary restart files (`LOUTRESTART`). The content of these files has been amended so that the restarted simulation will receive all 625 the relevant information. For restarting, set `IPIN=1` in the `COMMAND` file and rename or link the desired start-point file from `restart_XXX` file to `restart.bin` in the output directory. It is recommended to set `LOUTRESTART` to a relatively large value in order to minimise the computational and storage overhead.

### 9.4 Dynamical allocation of arrays

In FLEXPART 11 it is no longer necessary to specify dimensions of (nested) input fields, receptors, particle- or species-related 630 arrays at compile time by manually setting the dimensions in the `par_mod.f90` file. These arrays are now dynamically allocated as required by the size of the input data, enhancing user-friendliness. Another advantage is that it allows to containerise FLEXPART.



### 9.5 Updated SPECIES files

Together with the model code, a set of input files describing the physical-chemical behaviour for various species are provided
as templates (SPECIES_xxx, with xxx being a number in the model input directory); these properties can and should be set
by the users and are at their own discretion. Each file is dedicated to one 'species' which represents a real-world species or
group of species with a specific behaviour in the model. The information contained in that file is used to calculate wet and
dry deposition, gravitational settling, (radioactive) decay or destruction due to reaction with the hydroxyl radical, OH (OH
fields have to be prescribed). The parameter values in the template files provided contain suggested parameter values; short
references (name of the first author, year of publication or a web link) are provided to the sources from which the parameter
values were taken. A more detailed description is given in the Appendix (section+B).

### 10 Platform for interaction of users and developers

FLEXPART is a community model, allowing everyone to modify the code according to their specific needs, to fix bugs, add
functionality, and to communicate with the current developers. Since its inception, it has been available under the GPL V3
license as free and open software. A web-based project management and bug tracking system is available both for use by
developers and for interaction with users. With FLEXPART 11, this has been switched from a trac hosted by the Central
Institute of Meteorology and Geodynamics (now: Geosphere Austria) to a GitLab instance hosted at University of Vienna.
It includes a continuous integration (CI) functionality. The University of Vienna provides this open GitLab instance within the
PHAIDRA (Permanent Hosting, Archiving and Indexing of Digital Resources and Assets) service for long-term research data
storage (PHAIDRA, 2008). Currently, a few simple tests are bundled with FLEXPART, which are used in the CI. They should
be seen as a starting point for future enhancements. The continuous code testing supports community code development and
provides additional information on the deployment of FLEXPART under various scenarios (e.g. compilers and dependencies).
It further allows to provide a FLEXPART container, ready for deployment and easy porting of FLEXPART to various high-
performance computing environments or cloud instances.

### 655    11 Conclusions

We have presented the new version 11 of the FLEXPART model. A large number of development steps have been carried out,
which improve the accuracy, enhance computational properties, and add functionality.

   The following new features improve the accuracy of FLEXPART simulations:

1. The possibility of performing transport calculations in the native $\eta$ vertical coordinate system for ECMWF data, instead
of internally converting all fields to vertical coordinates in metres, results in both reduced absolute transport conservation
     errors of PV and other semi-conserved variables, and a better, but still not perfect, fulfilment of the well-mixed criterion
     in global domain-filling simulations.





2. An improved parameterisation scheme for wet scavenging of aerosols below clouds, an improved precipitation interpolation method, and a more transparent cloud layer identification method were introduced, resulting in the removal of artefacts that were present in previous model versions.

3. More accurate drag coefficients for aerosol particles of different sizes and shapes, which can be characterised in all three dimensions, in some cases dramatically improve the simulation of the atmospheric transport of coarse-mode aerosols.

4. The implementation of the optional linear chemistry module turns FLEXPART into a simple linear chemistry transport model. When activated, this makes it possible to initialise particle mixing ratios, incorporate the influence of surface fluxes and linear chemistry, and sample particle mixing ratios at receptor locations.

In terms of computational enhancements, the switch from MPI to OpenMP parallelisation brings better memory utilisation and improved scalability across a wide range of applications. However, we note that significant improvements could still be achieved by using co-arrays or a hybrid OpenMP-MPI implementation, especially in combination with a domain decomposition of the meteorological input data.

Finally, new use options were introduced in FLEXPART 11 and the SPECIES template files have now been documented. The option to replace the conventional particle release information (fixed release rates for a list of given spatio-temporal domains) with arbitrary initial particle conditions adds flexibility and, for certain cases, efficiency. The option for restarting a FLEXPART simulation after an unforeseen or planned termination has been made more complete and more computationally efficient. Particle information can now be dumped in a configurable manner in compressed NetCDF format.

Finally, to aid the user community, an instruction manual (https://flexpart.img.univie.ac.at/docs) has been written as a live document. A continuous integration environment based on `gitlab` is available for current developers and those who wish to contribute to the further development of FLEXPART.

*Code availability.* The code as described in this paper is available as a tarball from [https://gitlab.phaidra.org/flexpart/flexpart]. For future releases, and in order to obtain the latest version from the `git` repository, as well as bug reports and feature requests, please visit https://flexpart.img.univie.ac.at/, accessible also through the general FLEXPART home page https://flexpart.eu.

## Appendix A: Flexpart overview and code reorganisation

In FLEXPART, the modelling of computational particle movement involves a combination of interpolating properties from meteorological input data and computing properties that are intrinsic to each particle. This hybrid approach allows for a more comprehensive and realistic representation of particle behaviour in the atmosphere. Computational particles can represent one or more species, which may be gases or aerosols with certain properties. For brevity, computational particles are hereafter referred to as "particles".



This overview of the code is divided into three parts: meteorological input data used by FLEXPART (see section A1), computations related to the transport of particles (see section A2), and computations affecting the properties of particles (see section A3). We will only provide short summaries, since these parts have all been documented in publications accompanying previous releases (e. g., Stohl et al., 2005; Pisso et al., 2019) and are described in more detail in the manual: https://flexpart. img.univie.ac.at/docs. The main purpose here is to document how these computations are now organised within the source code, following its restructuring.

## A1 Input data

FLEXPART advances particles based on interpolated meteorological fields, namely grid-scale three-dimensional fields of wind velocities, density, temperature, specific humidity, cloud liquid water and ice content, as well as precipitation and various surface fields. In principle, any gridded data set could be used. Data formats and coordinate systems used as well as differences in the meteorological variables provided, however, would make a generic input interface rather complex. The main FLEXPART code described here supports two input formats, data from ECMWF's and from NCEP's forecast systems (IFS and GFS, respectively). For ECMWF data, the flex_extract software package (Tipka et al., 2020) is provided to extract, process, and store the required fields for use as FLEXPART input, including support for ECMWF's reanalyses. Note that in the case of ECMWF-based input, data on all model layers are used, whereas NCEP-based input comprises only pressure-level fields. Other formats could be read in by writing appropriate `gridcheck` and `readwind` subroutines and adding them to the `windfields_mod.f90` module. A full list of necessary input fields can be found in table A1. With the exception of the option of reading three precipitation fields per input time interval (see section 5), nothing has changed compared to FLEXPART 10.

Other variants of FLEXPART were developed to accommodate fields produced by specific meteorological models., e. g., FLEXPART-WRF (Brioude et al., 2013), FLEXPART-COSMO (Henne et al., 2016), FLEXPART-AROME (Verreyken et al., 2019), FLEXPART-HIRLAM (Foreback, 2023), and FLEXPART–NorESM/CAM (Cassiani et al., 2016). There also exists a FLEXPART version for very-high-resolution simulations (1 km), where turbulence parameterisations have been adapted to account for the fact that turbulence is partly already resolved by the meteorological input data (Katharopoulos et al., 2022).

## A2 Particle transport

The transport of particles is central to an LPDM and straightforward. FLEXPART advances particles using motion vectors. At each time step, these motion vectors combine the grid-scale wind velocity ($\bar{\mathbf{v}}$) from linearly interpolated meteorological input data, the parameterised turbulent wind velocity ($\mathbf{v}_t$, see section A2.2) and, for aerosols, the settling velocity ($\mathbf{v}_s$, see section A2.3). The resultant motion vector for that time step is $\mathbf{v} = \bar{\mathbf{v}} + \mathbf{v}_t + \mathbf{v}_s$. The new particle position is then given by

$$\mathbf{X}(t + \Delta t) = \mathbf{X}(t) + \mathbf{v}(\mathbf{X}, t)\Delta t. \tag{A1}$$

where $\Delta t$ is the internal time step of the model. In addition, particles may be displaced vertically due to convection (see section A2.1).





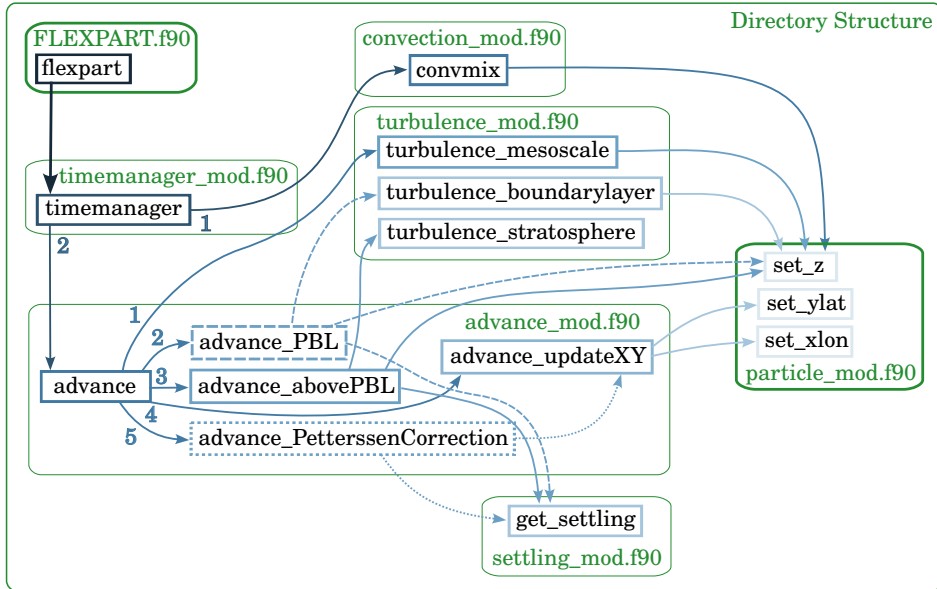

**Figure A1.** Simplified call tree for the transport of particles; green boxes represent modules containing subroutines (blue boxes) responsible for updating the location of particles through time.

As shown in figure A1, the time-manager module (`timemanager_mod.f90`) initiates, at each time step, the various

processes. Particles first undergo convection (if required), which only affects their vertical position, then the advance module (`advance_mod.f90`) is called, separately for each particle. Depending on the vertical position of the particle, different turbulence parameterisations are selected (see section A2.2). After having derived the displacement vector and obtained a first guess for the new location of the particle, one Petterssen correction step (Petterssen, 1940) is executed for greater numerical accuracy to move the particle to its final new position. This is a second-order numerical scheme, which is accurate enough for

the purpose of trajectory calculations given sufficiently short time steps (Seibert, 1993). Particle positions are updated within the particle module (`particle_mod.f90`), which is directly or indirectly called from the advance module.

**A2.1 Convection**

Vertical transport by moist, subgrid-scale convection is parameterised by using the scheme of Emanuel and Živković-Rothman (1999). In FLEXPART 11, all convection-related computations are located in the convection module (`conv_mod.f90`). The

735 steps required are handled by `convmix` and include the computation of the matrix describing the convective redistribution of mass in a grid column (`calcmatrix`), making use of to the convection scheme of Emanuel and Živković-Rothman (1999) (`convect` and `tlift`), followed by the actual convective displacement of particles in (`redist`). The sorting algorithm used in `convmix` has been changed to use the Fortran Standard Library (https://fortran-lang.org/), which codes the algorithm of Musser (1997). This new sorting algorithm is four orders of magnitudes faster; it is located in a separate module,

`sort_mod.f90`.



## A2.2 Turbulence

In FLEXPART 11, all turbulence-related computations have been organised in subroutines inside the turbulence module. This includes the turbulence computations inside of the atmospheric boundary layer (ABL), based on the Hanna and Chaughy 1982 scheme (Hanna, 1982), with added features from Ryall and Maryon (1998), and a skewed turbulence option by Cassiani et al. (2015), the turbulence above the ABL, a parameterisation based on pseudo-random number sampling and the length of the computational time step (Legras et al., 2003), and the mesoscale turbulence, parameterised using the method described by Maryon (1998), which uses the standard deviation of the wind velocities in the wind fields at the surrounding grid points. These turbulence schemes are called from the `advance module`, and can now be completely turned off by an option in the COMMAND file by setting `LTURBULENCE=0`. Mesoscale turbulence is switched off by default and can be switched on in addition by setting `LTURBULENCE_MESO=1`. Below, we give a short description of the different turbulence schemes and options that are currently implemented in FLEXPART.

*Turbulence in the atmospheric boundary layer.* Turbulence in the ABL is parameterised assuming a Markov chain based on the Langevin equation (Thomson, 1987). The Lagrangian time scales and turbulent velocity statistics for Gaussian turbulence are computed following either the scheme of Hanna (Hanna, 1982) or that of Ryall and Maryon (1998). Both distinguish neutral, stable and unstable conditions. Turbulent motions can either be calculated using FLEXPART's fixed synchronization time step, $\Delta t_s$, set by `LSYNCTIME` in the COMMAND file or use shorter time steps depending on the Lagrangian time scale $\Delta t_L$. The former option is computationally very efficient, but is inaccurate, as velocity auto-correlation cannot be taken into account if $\Delta t_s > \Delta t_L$. It is suitable only for long-range transport processes where details of turbulence in the ABL are less important. Adaptive time steps are enabled by `CTL>1` in the COMMAND; `CTL` determines by which factor the computational time step is kept shorter than $\Delta t_L$. For Gaussian turbulence, `CTL` values of at least 5 are recommended, and for the skewed turbulence scheme even larger values are needed. With CTL > 1, the time steps for the vertical component of turbulent modtion can be further refined by setting `IFINE>1`, with the value of `IFINE` determining the factor by which the time step is reduced.

FLEXPART offers the option to choose between Gaussian (Stohl and Thomson, 1999) and skewed turbulence (Cassiani et al., 2015). Gaussian turbulence is a suitable approximation for stable and neutral conditions. The error when using Gaussian turbulence under convective conditions is relatively small for sufficiently long transport distances where particles become well mixed throughout the ABL. Only for short-range (i.e. $\lesssim$1 h) dispersion in the convective ABL however, it is recommended to use the skewed turbulence scheme (Cassiani et al., 2015) by setting the switch `CBL` to 1 in the COMMAND file, as this scheme considerably increases computation time.

*Turbulence in the free troposphere and stratosphere.* Turbulent velocities above the ABL, both in the stratosphere and in the troposphere, are computed following Legras et al. (2003), using a constant vertical diffusivity ($D_z = 0.1\,\mathrm{m^2s^{-1}}$) to compute vertical turbulent velocities in the stratosphere, and a constant horizontal diffusivity ($D_h = 50\,\mathrm{m^2s^{-1}}$) to compute horizontal turbulent velocities in the free troposphere. These default values can be modified in the input files since version 10.4. A linear




transition layer of 1 km is used between the troposphere and stratosphere. The tropopause is defined as the first stable layer to fulfill the thermal tropopause criterion (i.e. the vertical temperature gradient is smaller than 0.002 km$^{-1}$). Standard deviations of each of the velocity components ($i = 1, \ldots, 3$) are then obtained as

$$\sigma_{v_i} = \sqrt{\frac{2D_i}{\Delta t}}.$$ (A2)

### A2.3 Gravitational settling

The gravitational settling of particles is called at each time step. The settling velocity of each particle is computed following Näslund and Thaning (1991), with a temperature dependent dynamic viscosity according to Sutherland (1893), and then added to the vertical motion. In FLEXPART 11, a settling module, settling_mod.f90, has been introduced containing all the relevant procedures. While previous versions of FLEXPART only considered spherical particles, FLEXPART 11 is able to calculate the settling velocity also for non-spherical particles and takes into account different types of particle orientations (see section 4). Gravitational settling is only calculated if a particle carries a single aerosol species with non-zero mass. This is necessary since aerosols of different size, shape or density exhibit different settling velocities; however, a particle can only follow a single trajectory.

### A3 Evolution of particle properties

In the previous subsection, we discussed how particles are advanced, using parameterisation schemes and meteorological data interpolated to the particle position. In addition, FLEXPART tracks certain properties of one or more so-called "species" for each individual particle. The main property considered is the "mass" of the particles. Upon release, particles are assigned a certain "mass", usually representing a fraction of the total mass to be released as specified in file RELEASES. In the most simple case, this "mass" is physical mass; if applied to radioactivty, it can be understood to represent activity. In the case of of backward simulations, it represents mass (or activity) mixing ratios (Seibert and Frank, 2004). Nevertheless, for simplicity, we often refer just to the mass (without quotes) of a particle.

During the simulation, FLEXPART takes into account various processes that may alter a particle's mass (or what "mass" actually represents) over time. For instance, particles can undergo dry and wet deposition, by which their mass is reduced, described in section A3.1 and section A3.2. Additionally, radioactive decay and chemical reaction with the hydroxyl radical are other important processes that lead to a reduction in particle mass over time (section A3.3). It should be noted that "particles" are hypothetical entities used for discretisation in the Lagrangian framework. They should not be considered as actual aerosol particles. By considering the evolving "mass" of the particles, FLEXPART can simulate the transport, dispersion, and fate of gases and aerosols in a realistic manner. It allows the model to account for the varying lifetimes of different substances based on their specific properties and the environmental conditions they encounter.





### A3.1 Dry deposition

Gases and aerosols can both undergo dry deposition, with the result of losing mass to the Earth's surface. The (dry) deposition velocity $v_d$ is calculated for a reference height ($h_{\text{ref}}$) of 15 m above ground. The value of $h_{\text{ref}}$ can be changed by the user, keeping in mind that it should not be too low, so that a statistically sufficient number of particles are considered, and not too high, as the calculation of dry deposition becomes inaccurate if $2\,h_{\text{ref}}$ falls outside the constant-flux layer. Dry deposition is applied for all particles below $2\,h_{\text{ref}}$ as

$$\Delta m = m(t) \left[ 1 - \exp\left( \frac{-v_d(h_{\text{ref}})\,\Delta t}{2\,h_{\text{ref}}} \right) \right], \tag{A3}$$

where $\Delta m$ is the amount of mass transferred from the particle to the dry deposition field. The deposition velocity is a function of height above ground and defined as

$$v_d(z) = -\frac{F_C}{C(z)}, \tag{A4}$$

where $F_C$ is the deposition flux to the surface, and $C(z)$ is the concentration of the species at height $z$. This deposition velocity
can be prescribed as a constant value (in the associated `SPECIES` file), or can be computed within FLEXPART using the resistance method as described in Wesely and Hicks (1977) for gases and Slinn (1982) for aerosols. For using the resistance method, certain chemical and / or physical properties of the species considered must be provided by the user in the respective `SPECIES` file. For a full description of the dry deposition scheme see Stohl et al. (2005).

In FLEXPART 11, all routines related to dry deposition are collected in `drydepo_mod.f90`. For aerosols, the module
now also considers new formulations of gravitational settling velocity for spherical and non-spherical particles as described in section 4.

### A3.2 Wet deposition

The wet deposition scheme in FLEXPART distinguishes between aerosols and gases, and between in-cloud and below-cloud scavenging. In FLEXPART 11, cloud parameters and precipitation are interpolated to the particle position, and the
scheme for below-cloud scavenging of aerosols has been replaced. Routines related to wet deposition are now collected in the `wetdepo_mod.f90` module. The revised wet deposition scheme is explained in more detail in section 5.

The in-cloud scavenging is based on the approach of Hertel et al. (1995) with advancements introduced by Grythe et al. (2017). The in-cloud scavenging coefficient, $\lambda_i$ ($\text{s}^{-1}$), is defined by (Hertel et al., 1995):

$$\lambda_i = \frac{S_i I}{H_i}, \tag{A5}$$

where $S_i$ is the dimensionless scavenging ratio between the concentration of a substance in precipitation and the concentration in air, $I$ is the precipitation rate ($\text{m s}^{-1}$), and $H_i$ is the cloud depth where the precipitation occurs (in m). For gases, $S_i$ is defined as

$$S_i = \left( \frac{1 - c_l}{H_{\text{eff}} R T} + c_l \right). \tag{A6}$$




Here, $H_{\text{eff}}$ is the Henry's law constant, $R$ is the perfect gas constant, and $T$ is the temperature. $c_l$ is the cloud liquid water

content expressed in $\text{m}^3$ of water per $\text{m}^3$ of cloud air, a dimensionless quantity.

For particles, $S_i$ is defined as

$$S_i = \frac{f_{\text{nuc}}}{c_l}, \tag{A7}$$

where $f_{\text{nuc}}$ is the fraction of aerosol that is activated. Grythe et al. (2017) introduced a new parameterisation for $f_{\text{nuc}}$ which

takes into account the different efficiencies of aerosols in acting as cloud condensation nuclei or ice nuclei.

In the original Hertel et al. (1995) work, $c_l$ was obtained from the purely empirical relationship,

$$c_l = 2 \cdot 10^{-7} \cdot I^{0.36}. \tag{A8}$$

This formulation is still used in FLEXPART when the cloud liquid water content $c_l$ is not available. Aside from the new

parameterisation of $f_{\text{nuc}}$, Grythe et al. (2017) also introduced an improved expression for $\lambda_i$ that can be expressed in terms of

$S_i$:

$$S_i = f_{\text{nuc}} \frac{H_i I \rho_{\text{water}}}{c_{\text{TWC}} I} r_{\text{icl}}, \tag{A9}$$

where $c_{\text{TWC}}$ is the cloud total water content ($\text{kg m}^{-2}$) and $I \rho_{\text{water}}$ the mass of water precipitating per unit time and area

($\text{kg s}^{-1} \text{m}^{-2}$). The dimensionless empirical constant $r_{\text{icl}}$ is introduced to account for the replenishment rate of cloud water dur-

ing precipitation and can be set in FLEXPART by `incloud_ratio` in `par_mod.f90`. This accounts for the replenishment

of cloud water from condensing water vapour (Grythe et al., 2017). Noting that the cloud water washout ratio ($R_w$ in $\text{s}^{-1}$) can

be defined as

$$R_w = \frac{I \rho_{\text{water}}}{c_{\text{TWC}}}, \tag{A10}$$

introducing the in-cloud replenishment correction, $r_{\text{icl}}$, gives

$$R_w = \frac{I \rho_{\text{water}}}{c_{\text{TWC}}} r_{\text{icl}}, \tag{A11}$$

and results in a decreased washout ratio (i.e., increased washout time scale) by $r_{\text{icl}} < 1$. We note that the current definition of

$r_{\text{icl}}$ (a dimensionless empirical parameter) is different from the original one used in Grythe et al. (2017), since they included the

density of water in the empirical constant that was therefore dimensional, i.e. $r_{\text{icl,Grythe}} = r_{\text{icl}} \rho_{\text{water}}$. This has now been changed

in the FLEXPART 11 code for clarity and readability. It also means that the value of $r_{\text{icl,Grythe}}$ as reported in Grythe et al.

(2017), 6.2, now corresponds to a of 0.0062 of the dimensionless $r_{\text{icl}}$. For completeness, as in Grythe et al. (2017), we rewrite

the in-cloud scavenging coefficient in terms of the new expression for $S_i$ directly as

$$\lambda_i = f_{\text{nuc}} \frac{I \rho_{\text{water}}}{c_{\text{TWC}}} r_{\text{icl}}. \tag{A12}$$

As a fallback in the case of lacking cloud water data, FLEXPART 11 uses the simple parameterisation for the total scavenging

which was used since the early versions of the model, and which is common for regulatory nuclear applications (BASE, 2012),

namely $\Lambda = A I^B$. Parameters $A$ and $B$ have to be provided in the `SPECIES` file as `weta`, `wetb`.





In previous FLEXPART versions, the precipitation rate was augmented on the base of a sub-grid scale parameterisation considering that only a fraction ($F < 1$) of a grid cell would actually experience precipitation (Stohl et al., 2005). It was based on horizontal resolution (150 km $\times$ 150 km) data (Hertel et al., 1995). As it is unclear which values would be appropriate for various finer grids of current or future meteorological models, we considered it justified to remove this parameterisation in FLEXPART 11. The wet scavenging in convective clouds will need to receive further attention in the future, ideally coupling it to the convection scheme.

### A3.3 Radioactive and chemical decay

FLEXPART is able to account for radioactive and/or chemical decay of particles by defining a half life $T_{1/2}$ (parameter `pdecay` $> 0$ in the corresponding `SPECIES` file; no decay if `pdecay` $< 0$). The decay affects the mass on particles travelling through the atmosphere as well as deposited mass as follows:

$$m(t + \Delta t) = m(t)e^{-\alpha\,\Delta t}, \tag{A13}$$

with $\alpha$ being the decay constant

$$\alpha = \frac{\ln 2}{T_{1/2}}. \tag{A14}$$

The treatment of radioactive and/or chemical decay remains unchanged compared previous versions. Decay is computed alongside dry and wet deposition and can be found in the corresponding modules. Decayed mass is subtracted from the mass of each particle in the time-manager module (`timemanager_mod.f90`).

### A3.4 Chemical reactions

With the introduction of the linear chemistry module into FLEXPART, simple linear reactions can generally be computed (e.g. OH, Cl reactions). Loss processes related to reactions with radicals are represented as a first-order, linear approximation in FLEXPART 11 — identical to FLEXPART 10.4 (Pisso et al., 2019) for OH, but now expanded to any linear reaction. The chemistry-related mass loss, $m$, is calculated as:

$$m(t + \Delta t) = m(t)e^{-\kappa\Delta t}, \tag{A15}$$

with $\Delta t$ being the reaction time step given by `lsynctime`. The temperature-dependent reaction rate $\kappa$ (s$^{-1}$) is then calculated as:

$$\kappa = CT^N e^{\frac{-D}{T}} c_{\text{reagent}}, \tag{A16}$$

where $C$, $N$ and $D$ are the species-specific constants defined in the `SPECIES` file (parameters `pcconst`, `pdconst`, and `pnconst`, respectively; turned off with negative values). $T$ is the absolute temperature, and $c_{\text{reagent}}$ the hourly concentration of the reagent (Atkinson, 1997).



The OH radical is the most important oxidant in the troposphere and although atmospheric chemistry can be highly non-linear, a first-order, linear loss approximation using prescribed OH fields is often still adequate, e. g., to simulate methane in the atmosphere (for FLEXPART 11: monthly averaged $3°$ x $5°$ OH fields with 17 vertical layers; following GEOS-Chem model by Bey et al. (2001) and read in from NetCDF files). Hourly OH variations are accounted for by modifying the monthly fields with the hourly photolysis rate of ozone $j$ based on a simple parameterisation for cloud-free conditions depending on the solar zenith angle:

$$OH = \frac{j}{j^*}OH^*, \tag{A17}$$

where $j^*$ and $OH^*$ are the monthly mean photolysis rate and OH concentration taken from the OH fields, respectively.

In order to be able to use chemistry fields with higher spatial and temporal resolution (as in, e. g., Fang et al., 2016, for OH), the user has to implement reading and utilising such data in the chemistry-related FLEXPART subroutines (`readreagents`, `getchemfield`, `readchemfield`, `getchemhourly`, and `chemreaction` located in the `chemistry_mod.f90` module).

## Appendix B: SPECIES files

Species in FLEXPART are either gases or particles (the case of air tracer can be considered as an inert gas); each category requires different parameters to be specified in the respective `SPECIES` file. Values of parameters which are not required are ignored. The parameter values contained in the template `SPECIES` files bundled with FLEXPART are listed in Table A3 for gaseous and in Table A4 for particulate substances.

### B1  Gaseous species

The parameters required for the dry deposition of gases are the inverse of the diffusivity relative to that of of $H_2O$ ($D$, `PRELDIFF`), the reactivity relative to that of $O_3$ ($f_0$, `PF0`, originally taken from Wesely (1989)), and Henry's constant (`PHENRY`, describing solubility). The relative diffusivity of gases not listed there (CO, $SO_2$, $CH_4$, $C_2H_6$, $PCB-28$, $\gamma-HCH$ (lindane), $N_2O$) could be calculated approximately using the square root of the ratio of molar weights between water and the respective species. The source for the relative reactivity $f_0$ is Table 2 in Wesely (1989), with values for CO and $CH_4$ taken from Clifton et al. (2022).

A collection of values of the Henry constant for many atmospheric compounds was recently compiled by Sander (2023); the list of values can also be found on-line at https://www.henrys-law.org/henry/. For most of the species, they are close to or identical to the previously used values from Wesely (1989). More significant differences were found for $HNO_3$, $SO_2$, $NH_3$ and $HNO_2$. Now, values from Sander (2023) haven been used except for the persistent organic pollutants, where well-established values exist from other sources: for PCB28 from Mackay et al. (2006) (as in FLEXPART 10.4), and for $\gamma$-HCH from Sahsuvar et al. (2003) (modified).





Concerning the parameters relevant for the wet deposition of gases, in-cloud scavenging depends on the Henry constant and the diffusivity, which were already discussed above. The values of `weta` for the coefficient of the simplified, fallback scavenging parameterisation were originally taken from Asman (1995) for gases. Now, some values have been modified as described in Table A3.

Radioactive noble gases are inert and do not undergo relevant wet or dry deposition. Depending on their half-life and the time scales under consideration, their decay may be considered. Otherwise they are to be treated like air tracers. Radioactive decay (for gaseous as well as particulate species) may either be simulated directly in the FLEXPART simulation, or it can be considered during post-processing. The latter approach may be useful if several nuclides with different half-lives but otherwise identical properties are simulated, so that not only CPU time but also memory and output file size can be reduced. In some cases, activities are desired for a given reference time – then it would also not make sense to include the decay in the simulation. For applying a decay correction to the mean concentration or deposition during an output time interval, for species simulated without decay, see Seibert et al. (2013). Half-life data of different nuclides can be obtained from the International Atomic Energy Agency's 'Live Chart of Nuclides' at https://www-nds.iaea.org/relnsd/vcharthtml/VChartHTML.html, or the U. S. National Nuclear Data Center at Brookhaven National Laboratory, https://www.nndc.bnl.gov/; half-life values given in days need to be multiplied with 86400 to obtain the value in seconds as needed for FLEXPART.

## B2   Particulate species

The most important parameter for both dry and wet deposition of particles is their diameter, or more properly, their size distribution, which in FLEXPART is assumed to be log-normal. Thus a mean geometric diameter and a logarithmic standard deviation have to be specified.

For the mean geometric particle diameter (`PDIA`) for accumulation-mode aerosol, such as ammonium nitrate or sulfate, as well as radionuclides bound to such aerosol particles, a typical value of $0.4\,\mu\mathrm{m}$ is proposed, as in previous versions. Note that this is a quantity that is variable according to the environmental conditions. We have also introduced a template for fine and coarse dust, with typical diameters of $0.2\,\mu\mathrm{m}$, and, respectively, $10\,\mu\mathrm{m}$.

Radioactive iodine as released from nuclear reactors in the case of accidents mainly consists of gaseous elemental ($I_2$). This gas tends to deposit on accumulation-mode aerosol particles. Some radioiodine may also be present as methyl iodide ($CH_3I$) (Nair et al., 2000). A new SPECIES file was created for methyl iodide, with wet deposition parameters of elemental and methyl iodide taken from Asman (1995) and Päsler-Sauer (2000). Concerning dry deposition, Henry's constant and reactivity data could be found in the literature, a mean value for the deposition velocity ($v_\mathrm{dep}$) of $0.1 \cdot 10^{-3}\ \mathrm{ms}^{-1}$ was introduced for methyl iodide (Müller and Pröhl, 1993). Particle-bound iodine behaves like other particle-bound nuclides (see below).

Concerning the wet deposition of particle-bound cesium-137, Van Leuven et al. (2023) have tried to adjust FLEXPART deposition parameters based on inverse modelling; however, as the wet deposition scheme has been changed from version 10.4 (used by these authors) to version 11, and the inverse method is also subject to significant uncertainties, we refrain from generally recommending their values.





**Table A1.** List of input variables

| Input field | Description | unit |
|---|---|---|
| **3D fields** | | |
| Horizontal velocities | grid scale velocities used for, e. g., particle displacement | m/s |
| Vertical velocity | Vertical velocities on model levels used for, e. g., particle displacement | etadot/Pa (IFS), m/s (GFS) |
| Temperature | Air temperature used for parameterisation schemes | K |
| Specific (IFS) or relative (GFS) humidity | Internal use of specific humidity for parameterisation schemes | kg/kg (IFS), % (GFS) |
| **2D fields** | | |
| Surface pressure | Pressure at the ground level | Pa |
| Snow depth | Thickness of snow layer used for dry deposition | m |
| Cloud cover | Fraction of the grid cell covered by clouds, used for wet deposition | $0-1$ |
| 10 metre horizontal velocity | Used to compute surface stress if not available | $\text{ms}^{-1}$ |
| 2 metre temperature | Used for parameterisation | K |
| 2 metre dew point (ECMWF only) | Used for parameterisation; for GFS, computed according to Bolton (1980) | K |
| Large-scale precipitation | Used in the wet scavenging scheme | $\text{mmh}^{-1}$ |
| Convective precipitation | Used in the wet scavenging scheme | $\text{mmh}^{-1}$ |
| Sensible heat flux (ECMWF only) | Used to compute Obukhov length; for GFS, computed using the profile method (Berkowicz and Prahm, 1982). | $\text{Jm}^2$ |
| Solar radiation (ECMWF only) | Used to calculate the surface resistance for gases; for GFS, solar radiation is assumed to be zero | $\text{Jm}^2$ |
| E-ward & N-ward turbulent surface stress (ECMWF only) | Surface stress used for dry deposition; for GFS, surface stress is calculated using Berkowicz and Prahm (1982). | $\text{Nm}^2 \text{ s}$ |
| Orography | Altitude of topography above sea level | m |
| Standard deviation of orography | Minimum mixing layer height to account for subgrid-scale variability | m |
| Land-sea mask | Over land: using default surface stresses; over sea: invoking surface stress computation including wind speeds | $0-1$ |
| Cloud liquid water content | Used in the wet scavenging scheme | kg/kg |
| Cloud ice water content | Used in the wet scavenging scheme | kg/kg |



**Table A2.** Set of equations to calculate the drag coefficient $C_d$ of aerosol particles for three types of orientation based on the shape correction scheme of Bagheri and Bonadonna (2016). $L$, $I$, and $S$ are longest, intermediate, and smallest axes of a particle, respectively. $Re$ is the Reynolds number, $k_S$ is Stokes' and $k_N$ is Newton's drag correction, $F_S$ is Stokes' form factor, $F_N$ is Newton's form factor, $d_{eq}$ is the diameter of a sphere of equivalent volume, and $\alpha$ and $\beta$ are empirical expressions. If non-spherical particles are assumed to be ellipsoids or fibers with $L/I > 20$, the term ($\frac{d_{eq}^3}{LIS}$) can be neglected for the calculation of $F_S$ and $F_N$ (i.e. $F_S = fe^{1.3}$, $F_N = f^2 e$). For more details see Bagheri and Bonadonna (2016).

| Random orientation | Maximum-drag orientation | Average orientation |
|---|---|---|
| $F_S = fe^{1.3}\left(\frac{d_{eq}^3}{LIS}\right)$ | $F_S = fe^{1.3}\left(\frac{d_{eq}^3}{LIS}\right)$ | $F_S = fe^{1.3}\left(\frac{d_{eq}^3}{LIS}\right)$ |
| $F_N = f^2 e\left(\frac{d_{eq}^3}{LIS}\right)$ | $F_N = f^2 e\left(\frac{d_{eq}^3}{LIS}\right)$ | $F_N = f^2 e\left(\frac{d_{eq}^3}{LIS}\right)$ |
| $\rho' = \frac{\rho_p}{\rho_f}$ | $\rho' = \frac{\rho_p}{\rho_f}$ | $\rho' = \frac{\rho_p}{\rho_f}$ |
| $\alpha = 0.45 + 10/(\exp(2.5\log\rho') + 30)$ | | |
| $\beta = 1 - 37/(\exp(3\log\rho') + 100)$ | | |
| $k_{S,rand} = (F_S^{1/3} + F_S^{-1/3})/2$ | $k_{S,hor} = 0.5(F_S^{0.05} + F_S^{-0.36})$ | $k_{S,average} = (k_{S,rand} + k_{S,hor})/2$ |
| $k_{N,rand} = 10^{\alpha[-\log(F_N)]^{\beta}}$ | $k_{N,hor} = 10^{0.77[-\log(F_N)]^{0.63}}$ | $k_{N,average} = (k_{N,rand} + k_{N,hor})/2$ |

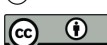


**Table A3.** List of gaseous FLEXPART species, including their file number, associated parameter values, and references, if available. $^{(*)}$ denotes values modified in FLEXPART 11, and $^{(**)}$ newly added species. T1/2 refers to the half-life; weta and wetb to below-cloud scavenging coefficients; reldiff, f0, and henry's constant to parameters for dry deposition; M to the molecular weight; and cconst, dconst, nconst to $C$, $D$ and $N$, respectively, used to calculate reaction rates (see Eq. A16).

| Species | T1/2 | weta | wetb | reldiff | henry | f0 | M | react. | cconst | dconst | nconst |
|---|---|---|---|---|---|---|---|---|---|---|---|
| Air tracer | NA | NA | NA | NA | NA | NA | 29 | NA | NA | NA | NA |
| $O_3$ | NA | 7.43E-5[1] | 0.62[1] | 1.6[2] | 1.0E-2[2] | 1[2] | 48 | NA | NA | NA | NA |
| NO | NA | 8.38E-5[1] | 0.62[1] | 1.3[2] | 2.0E-3[2] | 0[2] | 30 | NA | NA | NA | NA |
| $NO_2$ | NA | 7.12E-5[1] | 0.62[1] | 1.6[2] | 1.0E-2[2] | 0.1[2] | 46 | NA | NA | NA | NA |
| $HNO_3$[*] | NA | 5.82E-5[1] | 0.62[1] | 1.9[2] | 2.1E+5[11] | 0[2] | 63 | NA | NA | NA | NA |
| $HNO_2$[*] | NA | 7.04E-5[1] | 0.62[1] | 1.6[2] | 4.8E+1[11] | 0.1[2] | 47 | NA | NA | NA | NA |
| $H_2O_2$ | NA | 7.42E-5[1] | 0.62[1] | 1.4[2] | 1.0E+5[2] | 1[2] | 34 | NA | NA | NA | NA |
| $N_2O$[**] | NA | 7.22E-5[1] | 0.62[1] | 1.6[1] | 2.4E-2[11] | 0 | 44 | NA | NA | NA | NA |
| HCHO[**] | NA | 8.38E-5[1] | 0.62[1] | 1.3[2] | 6.0E+3[2] | 0[2] | 30 | NA | NA | NA | NA |
| PAN | NA | 5.04E-5[1] | 0.62[1] | 2.6[2] | 3.6E+0[2] | 0.1[2] | 121 | NA | NA | NA | NA |
| $NH_3$[*] | NA | 9.85E-5[1] | 0.62[1] | 1.0[2] | 5.9E+1[11] | 0[2] | 17 | NA | NA | NA | NA |
| CO[*] | NA | 8.62E-5[1] | 0.62[1] | 1.3[1] | 9.7E-4[11] | 0[3] | 28 | NA | NA | NA | NA |
| $SO_2$ | NA | 6.28E-5[1] | 0.62[1] | 1.9[2] | 1.3E+0[11] | 0[2] | 64 | NA | NA | NA | NA |
| $CH_4$[*] | NA | 9.31E-5[1] | 0.62[1] | 0.95[1] | 1.4E-3[11] | 0[3] | 16 | OH | 9.65E-20[4] | 1082[4] | 2.58[4] |
| $C_2H_6$[*] | NA | 8.38E-5[1] | 0.62[1] | 1.3[1] | 1.9E-3[11] | 0[3] | 30 | OH | 1.52E-19[4] | 498[4] | 2.00[4] |
| PCB-28 | NA | 3.99E-5[1] | 0.62[1] | 3.8[1] | 3.02E-3[5] | 0.1 | 258 | OH | 1.07E-11 | 1203 | 0.00 |
| $\gamma-$HCH[*] | NA | 3.86E-5[1] | 0.62[1] | 4.0[1] | 7.14E-2[6] | 0.1 | 291 | OH | 6.21E-11 | 1203 | 2.00 |
| $CH_3{}^{131}I$ | 6.934E+05 | 8.0E-7[1,10] | 0.62[1] | NA | NA | NA | NA | NA | NA | NA | NA |
| $^{131}I_2$[*] | 6.934E+05 | 8.0E-5[1,10] | 0.62[1] | 1.0E5 | 0.1 | NA | NA | NA | NA | NA | NA |

**Table A4.** List of particulate FLEXPART species, including their file number, associated parameter values, and references, if available. $^{(*)}$ denotes values modified in FLEXPART 11, and $^{(**)}$ newly added species. T1/2 refers to the half-life; crain_aero, csnow_aero are below-cloud scavenging efficiencies; ccn_aero, and in_aero are in-cloud scavenging parameters; density gives the density of an aerosol particle in $kg m^{-3}$; dia is the diameter of an aerosol particle in m; and dsigma its geometric standard deviation. A template for particulate radionuclides and an irregularly shaped particle are provided in the repository.

| Species | T1/2 | crain_aero | csnow_aero | ccn_aero | in_aero | density | dia | dsigma |
|---|---|---|---|---|---|---|---|---|
| Sulfate/nitrate ammonium[*] | NA | 1.0[7] | 1.0[7] | 0.9[7] | 0.1[7] | 2.00E+03 | 2.0E-7[8] | 1.5[8] |
| Black carbon[**] | NA | 1.0[7] | 1.0[7] | 0.9[7] | 0.1[7] | 2.00E+03 | 2.0E-8[8] | 1.5[8] |
| Mineral dust (fine)[**][7,9] | NA | 1.0 | 1.0 | 0.15 | 0.02 | 2.50E+03 | 2.20E-07 | 2.24 |
| Mineral dust (coarse)[**][7,9] | NA | 1.0 | 1.0 | 0.3 | 0.02 | 2.50E+03 | 1.23E-05 | 1.22 |

If no other values are known, weta $= 1E - 5$ amd wetb $= 0.8$ are recommended for aerosol particles BASE (2012).

[1] Asman (1995), [2] Wesely (1989), [3] Clifton et al. (2022), [4] Atkinson (1997), [5] Mackay et al. (2006), [6] Sahsuvar et al. (2003), [7] Grythe et al. (2017), [8] Tunved et al. (2013), [9] Groot Zwaaftink et al. (2022), [10] Päsler-Sauer (2000), [11] https://www.henrys-law.org/henry/



*Author contributions.*   Lucie Bakels led the code development, analysis and testing, contributed ideas, wrote the paper and created most figures. Daria Tatsii implemented the new settling scheme, performed testing, and wrote the gravitational settling section and created its figures. Anne Tipka, in collaboration with Petra Seibert, made improvements to the code, including the new wet deposition scheme. Rona Thompson implemented the LCM functionality, performed testing and wrote the linear chemistry module section. Marina Dütsch contributed to code development, bug fixing, and provided text and feedback. Michael Blaschek maintained version control, implemented developer tools,
and provided text. Petra Seibert came up with the idea of using hybrid coordinates (with Andreas Stohl), contributed to code development, and provided text and feedback. Katharina Baier participated in testing, discussions and provided feedback on the text. Silvia Bucci conducted testing, participated in discussions, especially on the new user functionality, and provided textual feedback. Massimo Cassiani contributed to code development, testing, and created the Fukushima section. Sabine Eckhardt contributed to code development, testing and provided the graphical abstract. Christine Groot Zwaaftink adapted FLEXDUST to FLEXPART 11, tested the LCM module, and provided textual
feedback. Stephan Henne did the initial OpenMP development and created the FLEXPART_CTM version that is now included in the Linear Chemistry Module in FLEXPART 11. Pirmin Kaufmann contributed to bug fixing and provided feedback on code development and text. Vincent Lechner contributed to the analysis of the density profiles and contributed to the text. Christian Maurer and Marie D. Mulder conducted and wrote the literature review on the FLEXPART SPECIES files. Ignacio Pisso debugged the GFS version of FLEXPART 11, merged the latest version of FLEXPART 10 with it, and contributed to testing and version control. Andreas Plach conducted testing of
backward trajectories, did code development, wrote part of the FLEXPART overview, and provided feedback. Rakesh Subramanian created the figure that shows the use of part_ic.nc for inverse satellite modelling. Martin Vojta performed testing for creating inverse modelling input and provided feedback on the text. Andreas Stohl provided ideas, suggestions for improving the code and contributed to the text.

*Competing interests.*   At least one of the (co-)authors is a member of the editorial board of Geoscientific Model Development.

*Acknowledgements.*   This study was supported by the Dr. Gottfried and Dr. Vera Weiss Science Foundation and the Austrian Science Fund
in the framework of the project P 34170-N, "A demonstration of a Lagrangian re-analysis (LARA)". R. Thompson and C. Groot Zwaaftink received financial support from the ReGAME project funded by the Research Council of Norway (grant no.325610). The computational results presented have been achieved [in part] using the Vienna Scientific Cluster (VSC). We acknowledge Christine Forsetlund Solbakken, NILU / https://www.colourbox.com/ for the creation of the graphical abstract. We acknowledge valuable input from Nikolaos Evangeliou, and Anjumol Raju. In addition, we acknowledge various public `python` packages that have benefited our study: `NumPy` (Harris et al., 2020)
, `Matplotlib` (Hunter, 2007), `Xarray` (Hoyer and Hamman, 2017), `SciPy` (Virtanen et al., 2020), and `cartopy` (Met Office, 2010 - 2015). We also acknowledge the use of scitools and the FORTRAN Documenter (FORD; https://github.com/Fortran-FOSS-Programmers/ford) in order to analyse the FLEXPART code.



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
