# Peer review of "FLEXPART version 11: Improved accuracy, efficiency, and flexibility"

_EGUsphere, 2024_

## Referee Comment (RC3)

**Review of "FLEXPART version 11: Improved accuracy, efficiency, and flexibility" by Bakels et al.**

The paper is a good summary of the improvements and updates made to FLEXPART at version 11, notably use of the native eta vertical coordinates with ECMWF meteorological data, accounting for the non-sphericity of particles, improvements to the wet deposition scheme, incorporation of a linear chemistry scheme, and the use of OpenMP parallelisation. The accuracy and performance of the model is also assessed using idealised tests, historic tracer experiments and more recent real-life events. The FLEXPART community should be commended on documenting and publication of the details of their model, including keeping this current and up-to-date. I find the paper well written and thorough and have only some minor comments and suggestions detailed below. I have also included a list of typographical errors I spotted. I recommend that the manuscript is accepted for publication after these queries and requests have been addressed.

1. Lines 46-48: The list of Lagrangian particle models seems to lack some of the key Lagrangian atmospheric dispersion models: MLDP0, NAME, SPRAY etc.
2. Line 48 "FLEXPART combines a unique set of capabilities no other model can offer..." and line 83 "offers many features not available in other models": the authors may want to rephrase these sentences, as it could be read as though the authors are implying FLEXPART is superior to other Lagrangian dispersion models. There is much commonality amongst Lagrangian particle dispersion models and many of the capabilities listed are present in other models. In addition, some of the functionality added to FLEXPART at version 11 has been present in other models for some time. Whilst it may well be true that FLEXPART is the only model to have all of the combination of functionalities listed, other Lagrangian models have some different functionality that FLEXPART may not (such as a Eulerian sub-grid model and radioactive decay into daughter products). The Lagrangian atmospheric dispersion modelling community benefits from different models and from the interactions within the community.
3. Line 92 refers to a Gitlab repository. Has this text been updated following comments by the editor on the suitability of Gitlab?
4. Line 101: The horizontal spatial resolution of the ERA5 meteorological data is quite coarse compared with the resolution of the ERA5 model (~30 km). Why is this? This seems particularly relevant given the use of the native vertical coordinate system to improve particle transport accuracy in this paper. If higher resolution meteorological data is available, this will also serve to improve particle transport accuracy.
5. Does the use of different vertical coordinate systems within and above the boundary layer (when using the eta option with ECMWF meteorological data) lead to any issues at the boundary layer top?
6. Figure 1 caption "absolute latitudes of 40 and 80°". I was uncertain what this meant here, although it was explained in the text. Could you say perhaps 'between 40 and 80 degrees north and between 40 and 80 degrees south' in the caption to be clearer?
7. Line 199: For readers not familiar with FLEXPART, could the options CTL and IFINE be defined?

8. Why is equation 5 not recovered, by setting kN and kS to be one in Equation 6? Should the two not be consistent in the limit of non-spherical → spherical?
9. CAPTEX results: I agree that there are no substantial differences (other than the NMSE and FOEX improvements), but the language is a bit inconsistent with that used to describe the ETEX results ("slightly better"). To be consistent and objective here, "slightly worse" would be more appropriate.
10. Table A1: "used for parameterisation". Which parameterisation?
11. Line 763: "with the value of IFINE determining the factor by which the time step is reduced". Is this 'further reduced'? In other words, is IFINE applied on top of CTL in the vertical?
12. Line 772-773: What about horizontal diffusivities in the stratosphere and vertical diffusivities in the troposphere? Are these assumed to be zero?
13. Line 912; "not listed there". It's not clear to me where 'there' is.
14. Is the data from the simulations being made available? Please check the journal requirements.

Typos:

1. Line 110: "employs a hybrid pressure-base vertical coordinates" could be "employs hybrid pressure-based vertical coordinates" or "employs a hybrid pressure-based vertical coordinate system".
2. Line 148. I don't think you want a 'respectively' here, as simulations for both heights were conducted with both vertical coordinate systems.
3. Line 194: Space required between 'by' and 'Cassiani'.
4. Figure 3 caption "are also reported near the top" should be "are also reported near the bottom"?
5. Line 297: "lead" should be "led"
6. Line 328: Can you have "stronger" precipitation, or should it be "heavier"?
7. Line 405: Should "were" be "was"?
8. Line 408: "8 and 20 meters" – above ground level, I presume but I'd prefer this to be clearly stated.
9. Line 412: "FA5" should be "FMS". SCC is also slightly worse for the eta coordinate, albeit comparable for the z coordinate.
10. Line 475: "starting" should be "start".
11. The legend and caption in Figure 8 do not agree on which are the solid, dashed and dotted lines.
12. Line 570: "That reduces" should be "This reduces".
13. Line 588: Remove "even".
14. Line 736: Remove 'to' - "making use of to the convection scheme" should be "making use of the convection scheme".
15. Line 737: Remove brackets around 'redist' – it is part of the sentence.
16. Line 762: "modtion" should be "motion"
17. Line 793: "of of" should be just "of".
18. Line 858: "now corresponds to a of 0.0062" should be "now corresponds to 0.0062". Is the mention of '6.2' on this line "the value of $r_{icl,Grythe}$ as reported in Grythe et al."? It wasn't clear to me.
19. Line 863: The use of a capital lambda for the scavenging coefficient, as opposed to a small lambda earlier could be confusing to the reader. Indeed, capital lambda is not defined.

20. Line 877" "compared previous versions" should be "compared to previous versions".
21. Lines 889-890: "parameters pcconst, pdconst, and pnconst, respectively" would imply C, D and N (in that order), which is not the order they appear listed on line 889.
22. Line 945: Requires an insertion of 'iodine' after 'gaseous elemental' or removal of the brackets around $I_2$.
23. There is some inconsistency in the formatting of units, with spaces missing between units in places (e.g., $ms^{-1}$ on line 949).
24. Tables A3 and A4 captions refer to the species file number, which I cannot find in the tables.

---

## Author Response (AR1)

**Reviewer 1**
* * *
GENERAL COMMENTS

This manuscript presents a new version of the Lagrangian particle dispersion model Flexpart. The main changes for this new version are described and relevant examples are given for several aspects of the model. The new version is compared with the previous version in terms of computational performance and accuracy by using observations obtained from two tracer experiments. As the Flexpart model is widely used by many researchers for several applications, the manuscript is of high interest to the community. The manuscript is well written and figures and tables are neat. The methods and assumptions are clearly described or properly referenced.

*We sincerely thank the reviewer, Pieter de Meutter, for taking the time to thoroughly review our manuscript. We appreciate your detailed feedback and constructive suggestions. We have carefully considered your comments and have made the necessary revisions to address the points you raised.*

*In the following, responses are in blue, and quoted text is show in green. Text after the little arrow '→' is newly introduced or modified manuscript text in the reaction to the reviewer's comments.*

SPECIFIC COMMENTS

l 90: Could the authors add instructions/hints to install Flexpart-11 and its dependencies (or refer to the online documentation)?

*We have clarified this by changing the following statement: "Accompanying this paper is a completely revised technical documentation of FLEXPART ([https://flexpart.img.univie.ac.at/docs](https://flexpart.img.univie.ac.at/docs))"*
*→ "Accompanying this paper is a completely revised technical documentation, including a download and installation guide, of FLEXPART (https://flexpart.img.univie.ac.at/docs)"*

l 95: "FLEXPART calculates particle trajectories using interpolated meteorological fields": the interpolation is performed in time and space?

*We clarify this: "FLEXPART calculates particle trajectories using meteorological data interpolated in time and space from time sequences of three-dimensional fields of, e.g., wind velocities, density, temperature, specific humidity, cloud liquid and ice water content."*

l 101: "All units are in International System of Units (SI) units, unless otherwise specified.": the context for this sentence is not very clear. Does it relate to the input meteorological data?

*We removed this sentence and report units throughout the text.*

l 149: "In addition, to avoid regions with low Coriolis force…": could the authors provide a motivation for avoiding regions with low Coriolis force?

*We changed the text as follows: → "We only used particles outside the subtropics and tropics, excluding the zone between 40° S and 40° N, as we expect the tracer conservation in this region to be worse in general, where the geostrophic balance is weak and deep convection is frequent."*

l 157: "N is the total number of particles in the sample": sample refers here to all particles in the domain between [-80, -40] and [40, 80] latitude?

*We rephrased the text as follows: "...where $y_i(t)$ and $y_i(t_0)$ are the quasi-conserved property of particle i at time t and $t_0$ of its trajectory, respectively, and N is the total number of particles in the sample." → "...where $y_i(t)$ and $y_i(t_0)$ is the quasi-conservative property of particle i at times t and $t_0$ of its trajectory, and N is the total number of particles in the domain between [80° S, 40° S] and [40° N, 80° N] latitude that fulfilled our selection criteria."*

Figure 1: can the authors provide an explanation why the improvement in the stratosphere is so large for potential temperature compared to the improvement for potential vorticity and specific humidity?

*It is true that the improvement for potential temperature in the stratosphere (61 %) is much bigger than that for absolute humidity (17 %) and PV (15 %), and also bigger than the improvement for all three quantities in the troposphere. The answer might lie in the way how errors in the vertical position (which is mainly affected by the new coordinate system) translate into errors of the quantity under consideration. Potential temperature gradients in the stratosphere are much larger than those of absolute humidity, and on top of that, absolute humidity is likely to be more uncertain already in the ECMWF analyses as values are so low. This does not explain why the relative improvement is also low for PV. Here, we have to search the reason in the large absolute values of PV especially in the upper stratosphere.*

Figure 2: the authors provide a number of possible causes for deviations from the well-mixed criterion. For |lat|>66° and between the surface up to +-1.5 km, the eta coordinates seem to result in a worse performance in terms of well-mixedness than the z coordinates. Do the authors have any possible explanation for this specific behavior?

*Unfortunately, we do not have an explanation. It is generally difficult to trace errors in well-mixedness to a specific reason. They are likely due to several different error sources (interpolation errors, numerical errors, mass consistency of driving ERA5 data) that partly lead to compensating effects. Furthermore, the deviations in well-mixedness may have their origin in completely different regions than where they materialize. The volume of the atmosphere where the z coordinates seem to lead to better well mixedness is quite small (polar regions below some 1.5 km), so we think this should not be overinterpreted.*

Figure 3:

- there is only one sentence of discussion for Figure 3, so I suggest to elaborate the discussion a bit or consider omitting the figure.

*Discussion of Figure 3 has been extended: '...for releases from Vienna, Austria, spherical particles with a diameter of 50 µm are deposited mainly in Central Europe, whereas volume equivalent fibres with an aspect ratio of 50 are also deposited in Eastern and Southern Europe (Figure 3a,b). The longer atmospheric transport of fibres results in an average horizontal travel distance 2.7 times greater than that of spherical particles (Figure 3c).'*

- the results shown in Figure 3 represent total deposition, that is, the combined effect of dry and wet deposition. Wet deposition will diminish the relative difference between the total deposition of spheres versus fibers? Therefore, it would be useful for the reader to have an idea of the amount of precipitation in January 2018 (that is, was it particularly dry or wet in that period?).

*In the case of the release shown in the 'Gravitational settling' section, more than 90% of the total deposition is dry deposition, which is shape-dependent. This is now mentioned in the caption of Figure 3: "In both cases, dry deposition accounts for more than 90% of the total deposition". Note that this specific behaviour is due to the large size of the released particles and their large settling velocities. For smaller particles (e.g., accumulation mode particles), wet deposition would dominate the total deposition.*

l 251: "Experiments show that non-spherical particles experience a larger drag in the atmosphere and therefore have lower settling velocities than spheres (Tatsii et al., 2024).": this is assuming identical particle volume and particle mass?

*Yes, identical particle volume and particle mass are assumed. The sentence has been changed accordingly: 'Experiments show that non-spherical particles experience a larger drag in the atmosphere and therefore have lower settling velocities than volume-equivalent spherical ones (Tatsii et al., 2024).'*

l 273: "PLA=2 PIA=2 PSA": do the authors mean "PLA=2, PIA=2 * PSA"?

*No, the sentence has been changed for clarification: '...characterized by PLA = 2 · PIA = 2 · PSA.'*

Figure 4: Comparison of panel (a) and (c) seems to suggest that in panel (c) wet deposition occurs even if there would be no rain in the meteorological data. Could the authors comment on this?

*Gaps between areas with deposition in (a) are mainly caused by the lack of interpolation for wet-deposition parameters, chiefly precipitation fields, in v10. The fact that the movement of*

*precipitation systems can be fast enough to skip grid cells within the 3 h interval between fields contributes as well. Therefore, it is (a) which is wrong, not (c). In addition, it is obvious that the pattern in (a) is unphysical whereas that in (c) looks realistic.*

Subsection 5.2: it is not clear to me whether the underlying problem with the interpolation relates to time, space or both. In the text, the focus seems to be on the temporal interpolation, while I thought the problem was more related to the fact that precipitation represents a grid box average value rather than a point value?

*The problem exists both in time and space. However, the problem in time is the bigger one, because*

*1) the central time for precipitation and the time for all other met fields do not match, and the workaround used up to now smoothes the precipitation time series, leading to a spread of precipitation into precipitation-free intervals (on top of what interpolation may do), as explained in Hittmeir et al. 2018.*

*2) Comparing spatio-temporal structures of precipitation fields and the actual spatial and temporal resolution of the met data shows that the difficulty to properly represent reality is greater in the temporal dimension.*

*We consider it desirable to extend the approach also to space, as briefly outlined in Hittmeir et al. 2018. However, this is not yet ready for implementation.*

In addition, could the authors briefly mention the two meteorological fields (large scale precipitation and convective precipitation?) that are used when numpf = 3?

*Large scale and convective precipitation are always read in. The two additional fields are rather additional in time and this refers to both LSP and CP fields. We clarified this in the following way: "This causes FLEXPART to read in two sets of precipitation fields (a set of three large scale and convective precipitation fields each, making it six fields in total) per input time interval instead of one. These additional fields represent two additional supporting time steps in between the original ones to represent precipitation as point values and, at the same time, preserve the integral precipitation in each time interval, guarantee continuity at interval boundaries, and maintain non-negativity (see Hittmeir et al., 2018, for more details)."*

Subsection 5.3: the fix for the presence of clouds in case of convective precipitation seems quite arbitrary? Could the authors give an idea of the sensitivity of the choices and whether they think this is a large source of model uncertainty or not? Lastly, is there a particular reason for starting the cloud at ground for convective precipitation above 0.1 mm/h?

*It is not totally arbitrary, it is based on some preliminary statistical investigations and meteorological experience.*

*Sensitivity will very much depend on the scenario, especially the height where particles are mainly located. As said, it is a preliminary, rough fix. Right now, we want to avoid missing wet deposition completely or to a large extent if there are no or only shallow resolved-scale clouds, and still do a little better than with a bulk formula as used up to v6.*

*We tested the following criteria for a global tropical release of 1 kg consisting of 1 million particles between 0 and 10 km spread between 1 January and 1 February 2017 using ERA5 data:*

1. *convp_precmin = 0.1 mm/h, low: 0-10km, high: 0.5-8km. The fix gets called in total 15% of all convective precipitation cases, of which 83% are classified as low and 17% as high, the total deposition is 0.8648 kg, a maximum value of 0.0371E-12 $kg/m^2$, and a mean of 0.0010952E-12 $kg/m^2$. (fig: orig.ps)*
2. *convp_precmin = 0.1 mm/h, low 0.1-10km, high: 1-8km. The total deposition is 0.8639E kg, a maximum value of 0.0364E-12 $kg/m^2$, and a mean of 0.0010941E-12 $kg/m^2$. (fig: l100h1000.ps)*
3. *convp_precmin = 1 mm/h, low 0.1-10km, high: 1-8km. As expected, the fix gets called again in total 15% of all convective precipitation cases, of which now 99.8% are low and 0.2% high, the total deposition is 0.8624 kg, a maximum value of 0.0357E-12 $kg/m^2$, and a mean of 0.00109225E-12 $kg/m^2$. (fig: l100h1000_1mm.ps)*
4. *convp_precmin = 5 mm/h, low 0.1-10km, high: 1-8km. The fix gets called again in total 15% of all convective precipitation cases, of which 99.9997% are low and 0.0003% high, the total deposition is 0.8623 kg, a maximum value of 0.0351E-12 $kg/m^2$, and a mean of 0.0010942E-12 $kg/m^2$. (fig: l100h1000_5mm.ps)*

*Attached are maps of the four cases, which show no visible differences. We therefore conclude that this is certainly something that needs to be further tested, especially on regional scale, but deposition seems to be not very sensitive to these criteria on a large scale. We added the following warning to the text:*

*"The results of preliminary tests indicate that the deposition resulting from convective clouds is not significantly influenced by the parameters used in this fix. However, further investigation is required to ascertain the full extent of the influence of these parameters and possible further optimisation."*

Figure 5: what is "sum" in the figure?

*"Sum" refers to the ratio of the sum of all cs137 and xe133 as defined by eq. 3 in Kristiansen et al. (2012). This gives a more robust measure of aerosol removal times than values taken from*

*individual stations (see Kristiansen et al., 2012). We added the following clarification to the caption: "Red boxes show the ratio of the sum of cesium-137 and the sum of xenon-133 over all stations."*

Subsection 5.4: could the authors provide motivation for aiming for an aerosol decay time of 9.3 d, close to the model median, rather than that based on the observations, which was 14.3 d?

*We did not aim for a specific aerosol lifetime, although we agree that the measurements would rather suggest a longer lifetime than the model. However, we do not want to tune the model to only this specific Fukushima case, since we do not know how representative it is for other weather situations or for other aerosol types. Most models have even shorter aerosol lifetimes than FLEXPART.*

l 368: "The mixing ratios for each species are calculated using the ratio of the mass of the species over the mass of air, where the mass of air is always carried by species number 1.": does this mean that the mass of air attributed to a particle changes during the simulation? If so, what processes in Flexpart modify the mass of air attributed to a particle?

*The mass of air represented by each particle is constant throughout the simulation. Only the mass of other species can vary (e.g. due to chemical reactions, radioactive decay, and fluxes at the surface). This is a special feature of the LCM mode, which is important there because particle densities are typically low in the domain-filling mode, which would lead to noisy results if mixing ratios were calculated by summing particle mass and dividing by air density and volume. Furthermore, LCM simulation times are often very long, and this method avoids problems with the slight violations of the well-mixedness condition in FLEXPART.*

l 713: "There also exists a FLEXPART version for very-high-resolution simulations (1 km), where turbulence parameterisations have been adapted to account for the fact that turbulence is partly already resolved by the meteorological input data (Katharopoulos et al., 2022).": this seems quite similar to Flexpart-AROME?

*There exist several FLEXPART versions for higher-resolution regional models (e.g., FLEXPART-AROME, FLEXPART-WRF, FLEXPART-ICON, etc.). However, to our knowledge only the Katharopoulos et al. (2022) paper describes a method to explicitly separate between resolved vs. unresolved turbulence.*

TECHNICAL CORRECTIONS

l 110: "The ECMWF's IFS employs a hybrid pressure-base vertical coordinates": omit "a"

*Corrected*

l 193: "byCassiani": add space

*Corrected*

Figure 3: in the caption: "top" should be "bottom"

*Corrected*

Figure 4: title in panel (c): consider writing numpf3 rather than npf3 to make it consistent with the text.

*We have made the title of panel (c) consistent with the text by replacing npf3 with numpf3.*

Table 1: "Number gives the total…": should be "n gives the total".

*Corrected*

l 413: "FA5" should be "FMS".

*Removed sentence in agreement with comments from the other reviewers*

Figure 8: the figure size should be smaller.

*It should have the width of a column in the published version. We will make it smaller for the next discussion upload.*

Table 2: In the caption, move the sentences "Weak scaling is defined by…" and "The serial comparison is done by …" to the text.

*Corrected*

l 511: omit "solid black lines" (should be in the figure caption or legend).

*Corrected*

l 512: omit "dashed black lines" (should be in the figure caption or legend).

*Corrected*

l 581: I suggest to omit "will".

*Corrected*

Figure 9: labels (a), (b) and (c) are missing.

*Corrected*

l 641: Should be "Appendix A".

*It should be Appendix B instead of Appendix (section+B), we have corrected this mistake. Thank you for pointing this out.*

l 793: omit the second "of".

*Corrected*

l 858: omit "a of".

*Corrected*

References:
- Hittmeir, S., Philipp, A., and Seibert, P.: A conservative reconstruction scheme for the interpolation of extensive quantities in the Lagrangian particle dispersion model FLEXPART, Geoscientific Model Development, 11, 2503–2523, https://doi.org/10.5194/gmd-11-2503-2018, 2018.
- Katharopoulos, I., Brunner, D., Emmenegger, L., Leuenberger, M., and Henne, S.: Lagrangian particle dispersion models in the Grey Zone of turbulence: Adaptations to FLEXPART-COSMO for simulations at 1 km grid resolution, Boundary-Layer Meteorology, 185, 129–160, https://doi.org/10.1007/s10546-022-00728-3, 2022.
- Kristiansen, N. I., Stohl, A., Olivié, D. J. L., Croft, B., Søvde, O. A., Klein, H., Christoudias, T., Kunkel, D., Leadbetter, S. J., Lee, Y. H., Zhang, K., Tsigaridis, K., Bergman, T., Evangeliou, N., Wang, H., Ma, P.-L., Easter, R. C., Rasch, P. J., Liu, X., Pitari, G., Di Genova, G., Zhao, S. Y., Balkanski, Y., Bauer, S. E., Faluvegi, G. S., Kokkola, H., Martin, R. V., Pierce, J. R., Schulz, M., Shindell, D., Tost, H., and Zhang, H.: Evaluation of observed and modelled aerosol lifetimes using radioactive tracers of opportunity and an ensemble of 19 global models, Atmospheric Chemistry and Physics, 16, 3525–3561, https://doi.org/10.5194/acp-16-3525-2016, publisher: Copernicus GmbH, 2016.
- Tatsii, D., Bucci, S., Bhowmick, T., Guettler, J., Bakels, L., Bagheri, G., and Stohl, A.: Shape Matters: Long-Range Transport of Microplastic Fibers in the Atmosphere, Environmental Science & Technology, 58, 671–682, https://doi.org/10.1021/acs.est.3c08209, pMID: 38150408, 2024.

**WD_spec001 (1e-12 kg m-2)**

[Figure]

longitude in degree east (degrees_east)

FLEXPART model output
Range of WD_spec001: 0 to 0.0371022 1e-12 kg m-2
Range of longitude in degree east: -179.25 to 180.25 degrees_east
Range of latitude in degree north: -89.75 to 89.75 degrees_north
Current nageclass: 0
Current time: 2.6784e+06 seconds since 2017-01-01 00:00
Frame 1 in File output_orig/grid_conc_20170101000000.nc

**WD_spec001 (1e-12 kg m-2)**

[Figure]

latitude in degree north (degrees_north)

longitude in degree east (degrees_east)

lucie Fri Aug  9 13:08:18 2024

FLEXPART model output

Range of WD_spec001: 0 to 0.0371022 1e-12 kg m-2

Range of longitude in degree east: -179.25 to 180.25 degrees_east

Range of latitude in degree north: -89.75 to 89.75 degrees_north

Current nageclass: 0

Current time: 2.6784e+06 seconds since 2017-01-01 00:00

Frame 1 in File output_100h1000l/grid_conc_20170101000000.nc

**WD_spec001 (1e-12 kg m-2)**

[Figure]

latitude in degree north (degrees_north)

longitude in degree east (degrees_east)

lucie Fri Aug 9 15:55:51 2024

FLEXPART model output
Range of WD_spec001: 0 to 0.0371022 1e-12 kg m-2
Range of longitude in degree east: -179.25 to 180.25 degrees_east
Range of latitude in degree north: -89.75 to 89.75 degrees_north
Current nageclass: 0
Current time: 2.6784e+06 seconds since 2017-01-01 00:00
Frame 1 in File output_100h1000l_1mm/grid_conc_20170101000000.nc

**WD_spec001 (1e-12 kg m-2)**

[Figure]

FLEXPART model output
Range of WD_spec001: 0 to 0.0371022 1e-12 kg m-2
Range of longitude in degree east: -179.25 to 180.25 degrees_east
Range of latitude in degree north: -89.75 to 89.75 degrees_north
Current nageclass: 0
Current time: 2.6784e+06 seconds since 2017-01-01 00:00
Frame 1 in File grid_conc_20170101000000.nc

**Reviewer 2**
* * *
The authors highlighted some recent changes in the FLEXPART model since the last similar publication for FLEXPART version 10.4 in 2019. The detailed descriptions of the changes are quite useful not only for the model users but also for other model developers. While the presentation is mostly clear, minor revisions are still needed. General and specific comments, as well as some editorial corrections are listed below.

*We would like to express our thanks to the reviewer for dedicating time and effort to thoroughly review our manuscript and appreciate your feedback and constructive suggestions. Please see below our responses to your comments.*

*In the following, responses are in blue, and quoted text is show in green. Text after the little arrow '→' is newly introduced or modified manuscript text in the reaction to the reviewer's comments.*

General comments:

The improved accuracy of the new version is not convincingly demonstrated from the examples in the manuscript. Although the semi-conserved property tests in section 3.2 show clear improvement of the new version, the tested properties are not exactly conserved as the authors also pointed out.

*Validation of trajectories is a notoriously difficult task, since there are few (if any) data sets that provide a solid ground truth against which calculated trajectories could be compared. For the mid- to upper troposphere, where we expect the largest improvements for FLEXPART 11, this is even more the case than for the boundary layer, where at least some tracer and constant-level balloon experiments are available. Other than the dynamical tracers which we use, we are not aware of any data set that could be used for that purpose. For instance, remote sensing data of volcanic ash plumes are unsuitable because the strength, height and time variation of emissions is never known accurately enough; the vertical motions of balloons do not follow vertical air motion closely enough; on-purpose tracer releases are not available.*

*The tracers which we use are indeed not perfectly conserved. We agree with the reviewer insofar as this cannot be used to assess the accuracy of individual trajectories. However, we believe that our method is valid for assessing the improvement of accuracy of the model, by comparing the results before and after the changes have been implemented, and by using a large set of trajectories. It is definitely expected that a better model should lead to overall better conservation*

*of these tracers, since model errors will add to physical (diabatic) effects leading to non-conservation. We show an improvement in accuracy, rather than an absolute measure of accuracy.*

*We measure the change in properties of particles over time of two runs with identical initial conditions and meteorological input data. The only difference between the two runs is the vertical interpolation of meteorological input data. Both runs show increasing deviations of the quasi-conservative properties over time from the perspective of the initial values. As the reviewer wrote, such a deviation is not necessarily incorrect. However, we can see from the figure in section 3.2 that the largest increments in the deviations occur when in the middle between the two hours where data is interpolated from, which points towards an increased change in quasi-conservative properties due to numerical and not due to physical reasons. Although not perfect, the simulation with the new scheme results in a smaller change in tracer values over time, which, in our opinion, can only be attributed to an improvement in the numerics of the model.*

The statistic results listed in Section 7 with tracer experiments show very marginal differences between the new and old versions. While Figure 6 compares the new model results with the ETEX measurements, it will be helpful to show the concentration fields predicted with the old version (or the difference between v 11 and v10.4). It will be beneficial to compare the vertical profiles of CAPTEX aircraft measurements and the predictions with both FLEXPART versions.

*As mentioned before, the biggest impact of the new scheme is expected in the stratosphere, and maybe upper troposphere over mountains. It is therefore not a surprise that the comparison with ETEX does not show a substantial improvement. Section 7 primarily serves to validate that we did not break something while making all the modifications to the code. We removed statements of getting better results for FP 11 as compared to FP 10 in this section, since we agree that these are marginal. Figure 6 for v10.4 output looks close to identical to the v11 output, which we added to the caption of figure 6:*
*"Note that results using z coordinates and FLEXPART 10.4 do not show significant differences."*

There is no doubt that the OpenMP parallelization implementation is important for the FLEXPART users, but many details of the technical aspects are probably not needed for a scientific paper. This reviewer suggest moving some of the contents in Section 8 to a supplementary material.

*We understand this argument. However, after thoughtful consideration, we prefer to keep the section within the main text as it is, because an article for GMD, as a journal specialised on describing model developments, should include the more technical model development aspects as well. We also noted that the other two reviewers did not raise this concern.*

Specific:

Lines 24-25, "... they can take into account all processes occurring during transport including nonlinear atmospheric chemistry": This gives an impression that Lagrangian models can not account for such processes. However, it is not true. Although it is convenient to use the Eulerian methods for such processes, the statement could be misleading.

*We corrected the statement in the following way: "The advantage of Eulerian methods is that they can take into account all processes occurring during transport including nonlinear atmospheric chemistry." → "Eulerian methods offer a convenient means of accounting for all processes that occur during transport, including nonlinear atmospheric chemistry."*

Line 48, "FLEXPART combines a unique set of capabilities no other model can offer, …":  Other Lagrangian 3D particle models have most if not all the capabilities listed here.  Thus it is not accurate. Please remove "no other model can offer".

*We removed the statement.*

Lines 149-150, "In addition, to avoid regions with low Coriolis force, we only used particles at latitudes north of 40◦N and south of -40◦S": Why? Can this be elaborated?

*We changed the text as follows: → "We only used particles outside the subtropics and tropics, excluding the zone between 40° S and 40° N, as we expect the tracer conservation in this region to be worse in general, where the geostrophic balance is weak and deep convection is frequent."*

Lines 411-412, "In fact, with the exception of FA5 and FOEX, all statistical values are slightly better for FLEXPART 11 than for FLEXPART 10": FLEXPART 11 using the z coordinate system actually has a better FA5 than FLEXPART 10.4. In addition, it is better to differentiate FLEXPART 10 and FLEXPART 10.4.

*Removed the line 'In fact, …' and changed all instances of FLEXPART 10 to FLEXPART 10.4.*

Line 425-426, "We also see no systematic large differences between FLEXPART 10.4 and FLEXPART 11, except for the NMSE values which again are better for FLEXPART 11":  The second part of the statement is not true.

*The NMSE is lower for FP 11 as compared to FP 10, both for ETEX and CAPTEX, but we removed the sentence since the differences between all individual statistical values are not significantly different.*

Line 421, "In table 1 we list the average and medians of …": What are actually listed in Table 1?

*Corrected, only means are listed.*

Line 445: Please clarify what "convection computations" mean here. Does that include the horizontal transport?

*We clarified this in the following way: "Convection computations" → "convection parameterisation (see section A2.1)"*

Line 559, "For the largest, this …": What does the largest refer to?

*We corrected this as follows: "For the largest, this..." → "For the largest problem size, this..."*

Line 610, "part_i.nc in the output directory": Should it be the input directory?

*We moved the part_ic.nc to the options directory, since this is indeed a more logical path than the output directory.*

Line 706, "…, whereas NCEP-based input comprises only pressure-level fields": Current GFS model has a hybrid sigma-pressure vertical coordinate.

*Yes, but unfortunately FLEXPART is built to ingest the NCEP pressure-level data. It would indeed be better to use the native sigma-level data. To our knowledge, nobody has adapted FLEXPART to use these data instead of the pressure level data.*

Line 776, 0.002 km^{-1}: Should the unit be K km^{-1}?

*Corrected*

Table A1, "E-ward & N-ward turbulent surface stress row": The unit of turbulent surface stress should be "N m^(-2) s" instead of "N m^2 s".

*Corrected*

Table A2: Please explain what α and β are.

*Explanations of α and β have been added:… and α and β are empirical expressions. → and α and β are sigmoidal functions of ρ'= ρp/ρf.*

Table A3: Units are needed for some of the parameters such as T1/2. In addition, it is better to have "1/2" as a subscript.

*Thank you for pointing this out. We have added the units and written ½ as a the subscript.*

Editorial:

List of affiliations are not in order.

*Corrected*

Line 193: "byCassiani"-> "by Cassiani"

*Corrected*

Line 471," Replace "printed" with "written" or "recorded".

*Corrected*

Line 735: Remove "to" after "making use of".

*Corrected*

Line 762, "turbulent modtion": Please correct the typo.

*Corrected*

Table A1: What is "etadot" in the Vertical velocity unit for IFS?

*Corrected to Pa/s, which is what FLEX_EXTRACT outputs after downloading the ECMWF data. See Tipka et al. (2020) for details of how vertical motion is handled.*

References:
- Tipka, A., Haimberger, L., and Seibert, P.: Flex_extract v7.1.2 – a software package to retrieve and prepare ECMWF data for use in FLEXPART, Geoscientific Model Development, 13, 5277–5310, https://doi.org/https://doi.org/10.5194/gmd-13-5277-2020, publisher: Copernicus GmbH, 2020.

**Reviewer 3**
* * *
The paper is a good summary of the improvements and updates made to FLEXPART at version 11, notably use of the native eta vertical coordinates with ECMWF meteorological data, accounting for the non-sphericity of particles, improvements to the wet deposition scheme, incorporation of a linear chemistry scheme, and the use of OpenMP parallelisation. The accuracy and performance of the model is also assessed using idealised tests, historic tracer experiments and more recent real-life events. The FLEXPART community should be commended on documenting and publication of the details of their model, including keeping this current and up-to-date. I find the paper well written and thorough and have only some minor comments and suggestions detailed below. I have also included a list of typographical errors I spotted. I recommend that the manuscript is accepted for publication after these queries and requests have been addressed.

*We are grateful to the reviewer, Helen Webster, for the time and effort spent on thoroughly reviewing our manuscript. We hope we have addressed your comments in a satisfactory manner and made the necessary revisions to improve the manuscript. Below, we listed our responses to your comments.*

*In the following, responses are in blue, and quoted text is show in green. Text after the little arrow '→' is newly introduced or modified manuscript text in the reaction to the reviewer's comments.*

1. Lines 46-48: The list of Lagrangian particle models seems to lack some of the key Lagrangian atmospheric dispersion models: MLDP0, NAME, SPRAY etc.

*Our apologies for this oversight, we included NAME, SPRAY, and MLDP0:*

*"Besides FLEXPART, several other LPDMs for regional and large-scale atmospheric transport modelling exist, e.g., HYSPLIT (Draxler et al., 1998), STILT (Lin et al., 2003), TRACMASS (Doos et al., 2017), MPTRAC (Hoffmann et al., 2022), and ATTILA (Brinkop et al., 2019)." → "Several other LPDMs exist, such as HYSPLIT (Draxler et al., 1998), STILT (Lin et al., 2003), TRACMASS (Doos et al., 2017), MPTRAC (Hoffmann et al., 2022), ATTILA (Brinkop et al., 2019), NAME (Jones et al., 2007), SPRAY (Tinarelli, et al., 2000) and the CMC's dispersion modelling suite (Damours et al., 2015)."*

2. Line 48 "FLEXPART combines a unique set of capabilities no other model can offer…" and line 83 "offers many features not available in other models": the authors may want to rephrase these sentences, as it could be read as though the authors are implying FLEXPART is superior to other Lagrangian dispersion models. There is much commonality amongst Lagrangian particle dispersion models and many of the capabilities listed are present in other models. In addition, some of the functionality added to FLEXPART at version 11 has been present in other models for some time. Whilst it may well be true that FLEXPART is the only model to have all of the combination of functionalities listed, other Lagrangian models have some different functionality that

FLEXPART may not (such as a Eulerian sub-grid model and radioactive decay into daughter products). The Lagrangian atmospheric dispersion modelling community benefits from different models and from the interactions within the community.

*We rephrased it to:*

*"FLEXPART combines the following capabilities: (i) a detailed..."*

3. Line 92 refers to a Gitlab repository. Has this text been updated following comments by the editor on the suitability of Gitlab?

*We added the following: "Accompanying this paper is a completely revised technical documentation, including a download and installation guide, of FLEXPART (https://flexpart.img.univie.ac.at/docs); a snapshot of the code used in this work is available at https://zenodo.org/records/12706633."*

4. Line 101: The horizontal spatial resolution of the ERA5 meteorological data is quite coarse compared with the resolution of the ERA5 model (~30 km). Why is this? This seems particularly relevant given the use of the native vertical coordinate system to improve particle transport accuracy in this paper. If higher resolution meteorological data is available, this will also serve to improve particle transport accuracy.

*We admit that data extracted at 0.25 (or 0.28125, to be more precise) deg resolution might have lead to slightly more accurate results, but at the cost of a quadrupled volume of input data. The 0.5 resolution data set is already very large, which is the reason for the choice made. Additionally, we had the impression the that horizontal variability of meteorological parameters at the smallest resolved scales in IFS output, including ERA5, is quite damped , and thus, extracting and using the data at the highest possible resolution may not result in a lot of improvement compared to 0.5 deg.*

*We added the missing vertical resolution to line 101: "For the examples provided in this paper, we use the most recent re-analysis dataset of ECMWF, ERA5 (Hersbach et al. 2020), with hourly 0.5°x 0.5° data on all of the 137 model levels as input to FLEXPART."*

5. Does the use of different vertical coordinate systems within and above the boundary layer (when using the eta option with ECMWF meteorological data) lead to any issues at the boundary layer top?

*We have not observed any issues at the boundary layer top. There is no signal visible at the ABL line in Figure 2, but since this is an average over a large area, possible isolated issues might not show up. However, we don't expect such problems, because FLEXPART first defines whether a particle is above or within the boundary layer, with subsequent use of the eta coordinate system if within. Then, it:*
*1) transforms the vertical position of the particle to the meter system,*
*2) computes the transport, including by turbulence, within the boundary layer in the meter system,*
*3) transforms the particle position and vertical velocity to eta coordinates*
*4) performs the Petterssen correction in eta coordinates*

6. Figure 1 caption "absolute latitudes of 40 and 80◦". I was uncertain what this meant here, although it was explained in the text. Could you say perhaps 'between 40 and 80 degrees north and between 40 and 80 degrees south' in the caption to be clearer?

*Adjusted, thank you.*

7. Line 199: For readers not familiar with FLEXPART, could the options CTL and IFINE be defined?

*We reworded the sentence:*

*"We used short time steps LSYNCTIME=600, with both the CTL and IFINE options set to 10) to increase the accuracy of turbulent transport in the ABL." → "We used short time steps, with the basic integration time step LSYNCTIME=600 s, and both the CTL and IFINE options set to 10 (thus allowing for reduced time steps in turbulence calculations, see section A2.2) to optimise the accuracy of the simulation of turbulent transport in the ABL."*

8. Why is equation 5 not recovered, by setting kN and kS to be one in Equation 6? Should the two not be consistent in the limit of non-spherical → spherical?

*Both equations are based on the generalized correlations for drag coefficient $C_D$ proposed by Haider and Levenspiel, 1989:*

$$C_D=(24/Re)*(1+C_1 Re C_2)+C_3/(1+C_4/Re)$$

*However, the two equations are not consistent in predicting the drag coefficients of both spherical and non-spherical particles. The fitting constants $C_1$, $C_2$, $C_3$ and $C_4$ are different in both cases. Equation 5, proposed by Clift and Gauvin, 1971, is based on empirical data of drag coefficients only for spherical particles (Figure 1 in Clift and Gauvin, 1971) and is not suitable for non-spherical particles, while Equation 6 is a semi-empirical model describing only non-spherical particles with different orientations of falling (Figure 23 in Bagheri and Bonadonna, 2016).*

9. CAPTEX results: I agree that there are no substantial differences (other than the NMSE and FOEX improvements), but the language is a bit inconsistent with that used to describe the ETEX results ("slightly better"). To be consistent and objective here, "slightly worse" would be more appropriate.

*This issue was also raised by reviewer #2, and we removed all mentions of 'better' and 'worse', since the differences are not significant.*

10. Table A1: "used for parameterisation". Which parameterisation?

*Added specifications for each field:*
*Temperature → Air temperature used in convection, chemical loss, and in the calculation of relative humidity*

*Specific (IFS) or relative (GFS) humidity → Specific humidity used in the calculation of convection, relative humidity, and dry air density*

*2 m temperature → Used to calculate friction velocity, Obukhov length, and convection*

*2 m dew point (ECMWF only) → Used to calculate friction velocity, Obukhov length, and convection*

11. Line 763: "with the value of IFINE determining the factor by which the time step is reduced". Is this 'further reduced'? In other words, is IFINE applied on top of CTL in the vertical?

*Yes, it is. The reason is that the shortest possible basic time step in FLEXPART is 1 second. However, applying IFINE allows to even use shorter time steps than 1 second for the vertical turbulent displacements. We clarified this in the text: "...with the value of IFINE determining the factor by which the time step is further reduced."*

12. Line 772-773: What about horizontal diffusivities in the stratosphere and vertical diffusivities in the troposphere? Are these assumed to be zero?

*Yes, at the moment they are assumed to be zero. This is indeed a topic where we might be able to improve FLEXPART in the future, e.g. by introducing a scheme similar to the one recently implemented in NAME (Mirza et al., 2024).*
*We added the following text to the paragraph in question: "Horizontal diffusion is neglected in the stratosphere, and vertical diffusion in the free troposphere. Already 20 years ago an attempt was made to include a CAT parameterisation in FLEXPART, but due to the difficulties inherent to this problem (CAT can be diagnosed only probabilistically, and it is hard to establish Lagrangian time scales for it) not pursued further. Recently work done for the NAME model (Mirza et al., 2024) may show a future way forward also for FLEXPART."*

13. Line 912; "not listed there". It's not clear to me where 'there' is.

*Removed "not listed there"*

14. Is the data from the simulations being made available? Please check the journal requirements.

*We added all the routines and a step-by-step guide on how to recreate the input data to our submission. These can be found in the Supplement. The data set is too large (>2 TB) for direct inclusion.*

Typos:

1. Line 110: "employs a hybrid pressure-base vertical coordinates" could be "employs hybrid pressure-based vertical coordinates" or "employs a hybrid pressure-based vertical coordinate system".

*Corrected*

2. Line 148. I don't think you want a 'respectively' here, as simulations for both heights were conducted with both vertical coordinate systems.

*Corrected*

3. Line 194: Space required between 'by' and 'Cassiani'.

*Corrected*

4. Figure 3 caption "are also reported near the top" should be "are also reported near the bottom"?

*Corrected*

5. Line 297: "lead" should be "led"

*Corrected*

6. Line 328: Can you have "stronger" precipitation, or should it be "heavier"?

*Corrected*

7. Line 405: Should "were" be "was"?

*Corrected*

8. Line 408: "8 and 20 meters" – above ground level, I presume but I'd prefer this to be clearly stated.

*Corrected*

9. Line 412: "FA5" should be "FMS". SCC is also slightly worse for the eta coordinate, albeit comparable for the z coordinate.

*Removed this sentence, since the differences are not significant.*

10. Line 475: "starting" should be "start".

*Corrected*

11. The legend and caption in Figure 8 do not agree on which are the solid, dashed and dotted lines.

*Corrected*

12. Line 570: "That reduces" should be "This reduces".

*Corrected*

13. Line 588: Remove "even".

*Corrected*

14. Line 736: Remove 'to' - "making use of to the convection scheme" should be "making use of the convection scheme".

*Corrected*

15. Line 737: Remove brackets around 'redist' – it is part of the sentence.

*Corrected*

16. Line 762: "modtion" should be "motion"

*Corrected*

17. Line 793: "of of" should be just "of".

*Corrected*

18. Line 858: "now corresponds to a of 0.0062" should be "now corresponds to 0.0062". Is the mention of '6.2' on this line "the value of ricl,Grythe as reported in Grythe et al."? It wasn't clear to me.

*We clarified it in the following way: "Removing the density of water from the empirical constant means that the value of $r_{icl,Grythe}$=6.2 kg/m³, as reported in Grythe et al. (2017), now corresponds to $r_{icl}$ = 0.0062 (dimensionless)."*

19. Line 863: The use of a capital lambda for the scavenging coefficient, as opposed to a small lambda earlier could be confusing to the reader. Indeed, capital lambda is not defined.

*The two different notations go back to those used in the original literature. Bulk formulations write \Lambda. We added the following clarification to the text: 'Note that we use \Lambda here instead of \lambda to be in agreement with previous literature referring to bulk values.'*

20. Line 877" "compared previous versions" should be "compared to previous versions".

*Corrected*

21. Lines 889-890: "parameters pcconst, pdconst, and pnconst, respectively" would imply C, D and N (in that order), which is not the order they appear listed on line 889.

*Corrected*

22. Line 945: Requires an insertion of 'iodine' after 'gaseous elemental' or removal of the brackets around I2.

*Corrected*

23. There is some inconsistency in the formatting of units, with spaces missing between units in places (e.g., ms-1 on line 949).

*Corrected and made the formatting consistent*

24. Tables A3 and A4 captions refer to the species file number, which I cannot find in the tables.

*Removed*

References:

- Brinkop, S. and Jöckel, P.: ATTILA 4.0: Lagrangian advective and convective transport of passive tracers within the ECHAM5/MESSy (2.53.0) chemistry–climate model, Geoscientific Model Development, 12, 1991–2008, https://doi.org/10.5194/gmd-12-1991-2019, 2019.
- D'Amours, R., Malo, A., Flesch, T., Wilson, J., Gauthier, J.-P., and Servranckx, R.: The Canadian

Meteorological Centre's Atmospheric Transport and Dispersion Modelling Suite, Atmosphere-Ocean, 53, 176–199, https://doi.org/10.1080/07055900.2014.1000260, 2015.

- Draxler, R. R. and Hess, G.: An overview of the HYSPLIT_4 modelling system for trajectories, Australian meteorological magazine, 47,295–308, https://www.researchgate.net/profile/G-Hess/publication/239061109_An_overview_of_the_HYSPLIT_4_modelling_system_for_trajectories/links/004635374253416d4e000000/An-overview-of-the-HYSPLIT-4-modelling-system-for-trajectories.pdf, 1998

- Döös, K., Jönsson, B., and Kjellsson, J.: Evaluation of oceanic and atmospheric trajectory schemes in the TRACMASS trajectory model v6.0, Geoscientific Model Development, 10, 1733–1749, https://doi.org/10.5194/gmd-10-1733-2017, 2017

- Grythe, H., Kristiansen, N. I., Groot Zwaaftink, C. D., Eckhardt, S., Ström, J., Tunved, P., Krejci, R., and Stohl, A.: A new aerosol wet removal scheme for the Lagrangian particle model FLEXPART v10, Geoscientific Model Development, 10, 1447–1466, https://doi.org/10.5194/gmd-10-1447-2017, 2017

- Hersbach, H., Bell, B., Berrisford, P., Hirahara, S., Horányi, A., Muñoz-Sabater, J., Nicolas, J., Peubey, C., Radu, R., Schepers, D., et al.: The ERA5 global reanalysis, Quarterly Journal of the Royal Meteorological Society, 146, 1999–2049, https://doi.org/10.1002/qj.3803, 2020

- Hoffmann, L., Baumeister, P. F., Cai, Z., Clemens, J., Griessbach, S., Günther, G., Heng, Y., Liu, M., Haghighi Mood, K., Stein, O., et al.: Massive-Parallel Trajectory Calculations version 2.2 (MPTRAC-2.2): Lagrangian transport simulations on graphics processing units (GPUs), Geoscientific Model Development, 15, 2731–2762, https://doi.org/10.5194/gmd-15-2731-2022, 2022

- Jones, A., Thomson, D., Hort, M., and Devenish, B.: The UK Met Office's next-generation atmospheric dispersion model, NAME III, in: Air pollution modeling and its application XVII, pp. 580–589, Springer, 2007

- Lin, J., Gerbig, C., Wofsy, S., Andrews, A., Daube, B., Davis, K., and Grainger, C.: A near-field tool for simulating the upstream influence of atmospheric observations: The Stochastic Time-Inverted Lagrangian Transport (STILT) model, Journal of Geophysical Research: Atmospheres, 108, https://doi.org/10.1029/2002JD003161, 2003.

- Mirza, A. K., Dacre, H. F., and Lo, C. H. B.: A case study analysis of the impact of a new free tropospheric turbulence scheme on the dispersion of an atmospheric tracer, Quarterly Journal of the Royal Meteorological Society, 150, 1907–1925, https://doi.org/https://doi.org/10.1002/qj.4681, 2024

- Tinarelli, G., Anfossi, D., Trini Castelli, S., Bider, M., and Ferrero, E.: A New High Performance Version of the Lagrangian Particle Dispersion Model Spray, Some Case Studies, pp. 499–507, Springer US, Boston, MA, https://doi.org/10.1007/978-1-4615-4153-0_51, 2000